# Democratic Training Against Universal Adversarial Perturbations

**Bing Sun**
Singapore Management University
`bing.sun.2020@phdcs.smu.edu.sg`

**Jun Sun**
Singapore Management University
`junsun@smu.edu.sg`

**Wei Zhao**
Singapore Management University
`wzhao@smu.edu.sg`

## Abstract

Despite their advances and success, real-world deep neural networks are known to be vulnerable to adversarial attacks. Universal adversarial perturbation, an input-agnostic attack, poses a serious threat for them to be deployed in security-sensitive systems. In this case, a single universal adversarial perturbation deceives the model on a range of clean inputs without requiring input-specific optimization, which makes it particularly threatening. In this work, we observe that universal adversarial perturbations usually lead to abnormal entropy spectrum in hidden layers, which suggests that the prediction is dominated by a small number of "feature" in such cases (rather than democratically by many features). Inspired by this, we propose an efficient yet effective defense method for mitigating UAPs called *Democratic Training* by performing entropy-based model enhancement to suppress the effect of the universal adversarial perturbations in a given model. *Democratic Training* is evaluated with 7 neural networks trained on 5 benchmark datasets and 5 types of state-of-the-art universal adversarial attack methods. The results show that it effectively reduces the attack success rate, improves model robustness and preserves the model accuracy on clean samples.

## 1 Introduction

Advances and success in deep learning have enabled the widespread use of Deep Neural Networks (DNNs) based machine learning models. DNNs become the algorithm of choice for a wide range of applications (Fu et al., 2016; Schroff et al., 2015; Bojarski et al., 2016; Vieira et al., 2017). However, despite their success, DNNs are found to make erroneous predictions when a carefully crafted, small magnitude human-imperceptible perturbation is added to an input (Goodfellow et al., 2015; Dong et al., 2018; Madry et al., 2018; Moosavi-Dezfooli et al., 2017). One can easily conduct adversarial attacks against the target network by generating adversarial examples utilizing such perturbations. The existence of adversarial examples has become a serious concern to systems based on DNNs especially in safety-critical applications. Neural network adversarial attacks can be input-specific (Goodfellow et al., 2015; Dong et al., 2018; Madry et al., 2018; Wang et al., 2021b; Zhang et al., 2022; Ganeshan et al., 2019) or input-agnostic (Moosavi-Dezfooli et al., 2017; Shafahi et al., 2020; Zhang et al., 2020b; Mopuri et al., 2018; Poursaeed et al., 2018; Hayes & Danezis, 2018; Liu et al., 2019). In the case of input-specific attacks or per-instance attacks, perturbations are individually optimized for each input to produce the corresponding adversarial example. In contrast, in input-agnostic attacks, a single perturbation is optimized for a set of inputs to produce an universal perturbation to generate a set of adversarial examples. Such perturbations are often referred to as universal adversarial perturbations (UAP), where the same perturbation applied to a range of clean inputs will cause the model to misclassify. Compared to input-specific adversarial attacks, UAPs could be considered more threatening since they are more efficient in terms of computation cost from the attack point of view. Furthermore, defending against UAPs poses a significant challenge, as it is hypothesized that they exploit and amplify legitimate features essential to the model's performance (Moosavi-Dezfooli et al., 2017; Zhang et al., 2020b; Borkar et al., 2020).

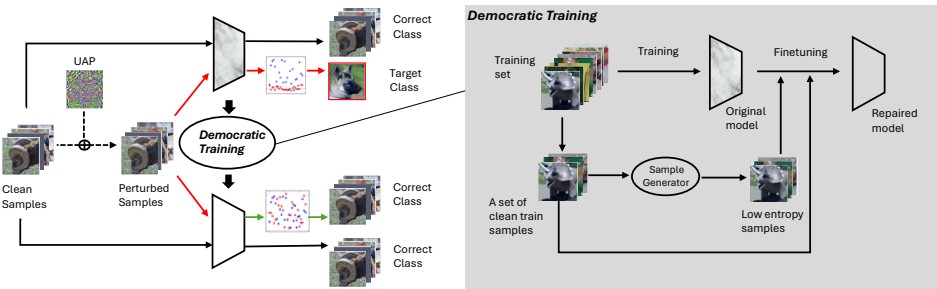

Figure 1: An overview of our framework

A number of practical and realistic attacks based on UAPs have been successfully conducted in various scenarios, i.e., image classification (Moosavi-Dezfooli et al., 2017), facial recognition (Sharif et al., 2016), object detection (Eykholt et al., 2018; Song et al., 2018), etc. Moosavi-Dezfooli et al. (2017) first explore the existence of UAPs. For a given set of training inputs, the proposed algorithm iteratively computes the perturbation to make an adversarial example across the decision boundary of the expected predicted category. Following this work, several different approaches have been proposed to generate UAPs from different aspects, utilizing different loss functions. These can be categorized into two main groups (Weng et al., 2024): 1) noise-based (Moosavi-Dezfooli et al., 2017; Zhang et al., 2020b;a; Mummadi et al., 2019), and 2) generator-based methods (Poursaeed et al., 2018; Mopuri et al., 2018; Naseer et al., 2021). Noise-based methods directly update the perturbation through optimization. On the other hand, generator-based methods train a generative network in prior to obtain the perturbation indirectly. UAP becomes a relevant threat in practice and it is important to manage such security risk and ensure neural networks are robust against such attacks. A range of existing works have been proposed to address the problem of defending machine learning models against UAPs. These include finetuning a given model's parameters with UAP perturbed samples (Moosavi-Dezfooli et al., 2017; Mummadi et al., 2019), inserting feature regeneration layers (Borkar et al., 2020), applying feature norm clipping techniques (Yu et al., 2021), etc. However, existing methods mainly focus on non-targeted attacks (Moosavi-Dezfooli et al., 2017; Shafahi et al., 2020; Benz et al., 2021) and often require to craft a large number of UAPs (Moosavi-Dezfooli et al., 2017; Mummadi et al., 2019; Shafahi et al., 2020; Benz et al., 2021; Borkar et al., 2020) or change the architecture of the original model (Borkar et al., 2020; Yu et al., 2021; 2023).

In this work, we focus on targeted universal adversarial attack which is both more relevant from an attacker point of view (i.e., so that the attacker can trigger specific target outcome) and more challenging from a defender point of view. Our approach does not require constructing UAPs or modifying the model architecture. We propose a scalable algorithm that mitigates the effect of UAPs through entropy based model enhancement. Specifically, as described in Figure 1 we propose *Democratic Training* with the key idea of enhancing a given neural network by adjusting the weights of hidden neurons towards the correct predictions in the presence of UAPs. We first analyze the distribution of hidden neuron activation when an input perturbed with UAP is supplied to a model and compare that with the activation when clean samples are supplied. We study the entropy of such hidden neuron activation and our empirical results suggest that the presence of UAP causes layer-wise entropy to drop and such effect becomes more severe at deeper layers. We conjecture that this is because the UAP enforces the "power" of certain features, which subsequently dominates the prediction. Base on such result, we propose to mitigate the effect of UAPs through adversarial finetuning guided by hidden layer entropy, or philosophically speaking, enforcing democracy in the decision making. We compare the performance of our work with existing solutions on UAP defense and show that *Democratic Training* improves existing approaches significantly.

## 2 PRELIMINARIES

### 2.1 UNIVERSAL ADVERSARIAL PERTURBATION

We start with introducing the notation for targeted UAP attacks. Given a trained neural network $N$, a test dataset $X$ and let $y_t$ represent the attacker-chosen target class. A targeted UAP is a perturbation

$\delta$ that satisfies the following:

$$N(x + \delta) = y_t$$
$$\| \delta \|_p \leq \epsilon \qquad (1)$$

where $x \in X$ and $|X|$ is sufficiently large and $\delta$ is bounded by certain $l_p$ norm ($\| \delta \|_p \leq \epsilon$). We remark that here we focus on a definition of vicinity based on $l_p$ norm. In general, it can be defined in other forms as well. For instance, a UAP can take the form of a patch that is small in size but applying it to a range of clean inputs, the model will classify the perturbed inputs as the target class.

The existence of UAP shows that there are systemic vulnerabilities in the model which can be exploited by an attacker regardless of the input. Hence, UAP attacks pose serious threats in real-world applications of neural networks such as attacking facial recognition systems where incorrect identity is returned (Amada et al., 2021; Zolfi et al., 2022), autonomous driving systems where a wrong traffic sign or road condition is misidentified (Benz et al., 2020; Eykholt et al., 2018), speech recognition systems which may cause various systems to interpret human commands wrongly (Sun et al., 2024), malware detection systems where suspicious programs may bypass the detection (Castro et al., 2021) and many others (Moosavi-Dezfooli et al., 2017; Zhang et al., 2020b; Metzen et al., 2017; Wallace et al., 2019).

## 2.2 EVALUATION METRICS

**Attack Success Rate (SR)**: This metric measures the percentage of adversarial samples (except the samples of the target class $y_t$) classified to the target class $y_t$:

$$SR = \sum_{x \in (X - X_t)} \frac{|N(x + \delta) = y_t|}{|X| - |X_t|} \qquad (2)$$

where $x \in X$ represents a clean input from dataset $X$, $X_t \subset X$ represents a set of samples from the target class $y_t$.

**Adversarial Accuracy (AAcc.)**: This metric measures the accuracy of adversarial examples (where $y_x$ represents the label of sample $x$):

$$AAcc. = \sum_{x \in X} \frac{|N(x + \delta) = y_x|}{|X|} \qquad (3)$$

## 2.3 ENTROPY

**Shannon Entropy:** In information theory, the entropy of a random variable represents the average amount of "information" or "uncertainty" associated with the variable's possible outcomes. The concept of information entropy was introduced by Claude Shannon (Shin & Kim, 1949), where the Shannon entropy is proposed to quantify the amount of information carried by a variable. For a random variable $v$, which takes values from the set $V$ that follows the probability distribution $p : V \to [0, 1]$ the entropy of $v$ is defined as:

$$H(v) = - \sum_{v \in V} p(v) \log p(v) \qquad (4)$$

where the summation denotes the sum over the variable's possible values.

**Measure the Entropy of A Neural Network:** The concept of entropy can be applied in neural networks for different purposes. Appendix 8.2 shows two methods proposed in existing works measuring the entropy of a given neural network. In this work, we propose to measure layer-wise entropy to understand how UAP fools a given neural network. The details are provided in Section 3.1.

## 2.4 THREAT MODEL

Our approach aims to mitigate the effect of UAPs for third-party trained neural networks. In this work, we assume an evasion threat where a set of clean data is available to the adversary.

- *Adversary goals.* The goal of the adversary is to generate UAPs such that once applied to a range of clean inputs, the model will classify the perturbed inputs wrongly.

- *Adversarial capabilities.* We assume the adversary has white-box access to the model and is capable of crafting UAPs.

- *Adversarial knowledge.* We assume that the adversary has the information on the target model's architecture, inner parameters and optimization algorithms.

Our goal is to mitigate the effect of UAPs on a given model with minimum assumptions. Specifically, we assume the defender has the following knowledge about the neural network:

- *Defense goals.* We aim to design a strategy that can remove the effect of UAPs from the model by adjusting the model parameters.

- *Defender's capabilities.* We assume the defender has white-box access to the neural network model. The defender has information about the model architecture but cannot interfere with the training process.

- *Defender's knowledge.* We assume a small set of clean data is available (as it is usually the case in practice), either given by the model provider or collected by the defender, to test the model's performance.

## 2.5 OUR PROBLEM

**Problem.** *Let $N$ be a neural network which is assumed to be obtained from a third party; $x$ is an input and $\epsilon$ is a small positive threshold. The UAP defense problem is to mitigate the effect of UAPs on $N$ such that the predictions of inputs patched with UAPs stay robust. Furthermore, the UAPs are bounded by $l_p$ norm where $\| \delta \|_p \leq \epsilon$. We would also require that the model's performance on clean data is minimally affected after the mitigation process.*

## 3 OUR APPROACH

To understand how UAPs deceive a model, we first conduct a systematic analysis of model behaviors from the lens of entropy. We study layer-wise entropy of a given model with and without the presence of UAPs. As we shall show in Section 3.2, the presence of UAPs will cause the layer-wise entropy to be abnormally lower than that on clean inputs. Furthermore, such effect becomes more severe at deeper layers. Based on these findings, we propose *Democratic Training* which conducts entropy-based model enhancement to repair the given model such that the effect of UAPs is mitigated.

## 3.1 ENTROPY MEASUREMENT

Firstly, we present how entropy is measured in this work. For a given neural network $N$, consisting of $n$ layers, we treat each layer $l$ as a single random variable, characterized by its input $x_l$ and output $x_{l+1}$. Thus, for a layer $l$ containing $d_l$ neurons, given an input to this layer $x_l = \{x_l^0, \ x_l^1, \ \cdots, \ x_l^{d_{l-1}-1}\}$, its layer-wise entropy is calculated as:

$$
\chi_l = \sigma(W_l x_l + b_l)
$$
$$
p_l = softmax(\chi_l)
$$
$$
H_l = - \sum_{k=0}^{k=d_l-1} p_l(k) \log p_l(k)
$$

(5)

where $W_l$ and $b_l$ are the weight and bias parameters of layer $l$ and $\sigma$ is the activation function of layer $l$. Intuitively, we treat the activated value $p_l(k)$ of each neuron in layer $l$ as the activation probability for neuron $k$, and calculate the Shannon entropy of $p_l$ following Equation 4. For a given input to layer $l$, higher layer entropy $H_l$ indicates higher ambiguity and lower entropy $H_l$ indicates higher certainty.

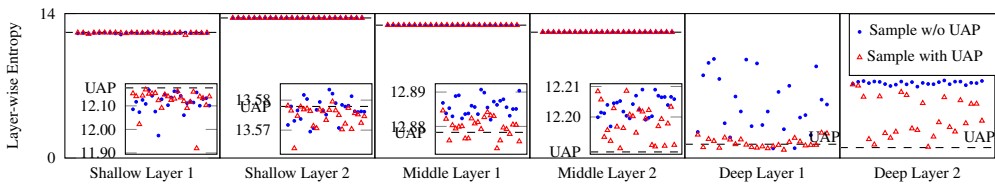

Figure 2: Layer-wise entropy. Enlarged view for shallow and middle layers is provided.

## 3.2 ENTROPY ANALYSIS

To understand how UAP fools a trained neural network, we conduct empirical study on the layer-wise entropy of the model as follows:

- *Step 1.* Given a pretrained neural network, generate a UAP such that the model classifies samples perturbed with the UAP as the target class.

- *Step 2.* Analyze the layer-wise entropy with clean samples only, i.e., we randomly select a set of clean samples and calculate their layer-wise entropy defined in Equation 5.

- *Step 3.* Apply the UAP generated in *Step 1* to the same set of clean samples selected in *Step 2* and analyze the layer-wise entropy.

- *Step 4.* Take the UAP itself as an input to the model and analyze the layer-wise entropy.

We conduct the analysis on all models shown in Table 1 and for each model, multiple targeted UAPs are generated using a state-of-the-art UAP attack method DF-UAP (Zhang et al., 2020b). We study the entropy of the pooling layers and the last layer of each stage. We observe similar results across all models and we show the results on $NN_1$ for shallow 1 (max pooling layer of stage 1), shallow 2 (end of stage 2), middle 1 (end of stage 3), middle 2 (end of stage 4), deep 1 (end of stage 5) and deep 2 (last average pooling layer) layers for illustration purpose. As shown in Figure 2, at shallow layers, the entropy spectrum for clean and UAP infected samples are quite similar. At middle layers, the entropy of some UAP infected samples becomes smaller than that of the clean sample, but there is no clear boundary for separating the two for all samples. At deep layers, we observe clear separation between entropy from clean and perturbed samples, where UAP infected samples show abnormally small entropy compared to that of clean samples. These results suggest that the presence of UAP will cause layer-wise entropy to drop and such effect becomes more severe at deeper layers. We interpret entropy as an indicator of the neural network's uncertainty in classifying the intermediate features. High entropy suggests the features are ambiguous while low entropy indicates the model is more certain on classifying the features. Our analysis results show that the presence of UAP will cause the layer-wise entropy to drop significantly, and such lower entropy indicates the model is more certain on its classification at the same layer. As shown in Figure 2, the entropy distribution of UAP perturbed samples leans towards the entropy of the UAP, i.e., the UAP dominates layer-wise entropy rather than the original image. At deeper layers, the entropy of UAP itself drops and is much lower than the entropy of clean samples, while the entropy of UAP perturbed samples follow such trend closely. We believe that, UAPs contain dominant features that cause the model to be certain on the prediction class at earlier layers, i.e., instead of features from the original sample, features from UAP lead the model to predict the target class. Similar findings are reported in existing work (Zhang et al., 2020b) that suggests UAPs contain dominant features and original images behave like noise to them. We argue that such dominant features cause the layer-wise entropy to drop which dominates the model prediction.

## 3.3 ENTROPY-BASED REPAIR

Based on our analysis results presented in Section 3.2, we design a general approach for mitigating UAPs called *Democratic Training*, which aims to finetune the model such that it learns to predict low-entropy samples (by effectively reducing the presence of certain dominate features in these samples). During this process, we introduce a *Sample Generator* that will craft low entropy samples from clean samples to mimic the effect of UAPs and guide the model to the correct prediction. Note that, the *Sample Generator* does not require information about the attack target class unlike existing

**Algorithm 1:** $Remove(I, N, m, \epsilon)$

1  **for** *n epochs* **do**
2      **for** *each batch b* **do**
3          $I_b^{en} \leftarrow$  $SampleGenerator(I_b, N, m, \epsilon)$;
4          $\mathbf{L}(i, i_{en}) = \alpha \mathbf{L}_{cce}(i_{en}) + (1 - \alpha)\mathbf{L}_{cce}(i)$;
5          $\mathbf{J}(\theta) = \frac{\partial \mathbf{L}(.)}{\partial \theta}$;
6          $\theta \leftarrow \theta - \gamma_\theta \cdot SGD(\mathbf{J}(\theta))$;

7  **return** $\theta$;

**Algorithm 2:** $SampleGenerator(I, N, m, \epsilon)$

1  **for** *m iterations* **do**
2      $\mathbf{L}(i) = \mathbf{H}(i)$;
3      $i \leftarrow i + \frac{\epsilon}{4} \cdot sign(\nabla_i \mathbf{L}(i))$;
4      $i = Clamp(i, \epsilon)$;

5  **return** $i$;

works that rely on pre-computed perturbations (Moosavi-Dezfooli et al., 2017). As described in Algorithm 1, *Democratic Training* requires a small set of clean sample $i \in I$ ($\leq 5\%$ of training set) to finetune the original model $N$. For each epoch and each batch during finetuning, the *Sample Generator* transforms a batch of clean inputs ($I_b$) into low entropy samples ($I_b^{en}$) as described in Algorithm 2. Starting from clean sample $i$, the perturbation is updated based on the projection of the layer-wise entropy loss ($\mathbf{H}(i) = -H_l(i)$) iteratively. At each step, a Clamp operation is applied to the perturbed sample to keep it within the perturbation bound. Next, *Democratic Training* calculates the loss of clean and low entropy samples as below:

$$\mathbf{L}(i, i_{en}) = \alpha \mathbf{L}_{cce}(i_{en}) + (1 - \alpha)\mathbf{L}_{cce}(i) \tag{6}$$

where $\mathbf{L}_{cce}$ represents the categorical cross entropy loss, $i$ represents a clean sample and $i_{en}$ represents a low entropy example generated. In Equation 6, $\alpha \mathbf{L}_{cce}(i_{en})$ aims to guide low entropy samples towards the correct prediction by minimizing their cross entropy loss and $(1 - \alpha)\mathbf{L}_{cce}(i)$ aims to keep the loss on clean samples low. Parameter $\alpha \in (0, 1)$ controls the trade-off between the effectiveness of UAP removal and performance on unperturbed inputs during the optimization process. To make sure the loss on the low-entropy samples is low, the model must learn to ignore those dominating features present in the low-entropy samples, i.e., learn to predict based on many features rather than a small number of dominating features. In *Democratic Training*, Back-propagation is adopted using the projected gradient descent (PGD) method (Madry et al., 2018). Finally, *Democratic Training* returns the updated model parameter $\theta$ as the result. Different from exiting methods (either generate UAPs in prior or on-the-fly), *Democratic Training* does not rely on generating UAPs and are thus not limited to specific UAP attacks.

The overall time complexity of Algorithm 1 is $\mathbf{O}(n \cdot m \cdot |I|)$, where $|I|$ is the size of the clean dataset used, $n$ is the number of epochs to finetune and $m$ represents number of iterations required to generate low entropy samples. Although *Democratic Training* requires multiple iterations to transform clean samples into low entropy samples, converting clean samples into low entropy samples is much simpler than generating UAPs and the total amount of samples to transform ($n \cdot |I|$) does not depends on the number of classes in a given dataset since the *Sample Generator* does not require any information on the target class. This is a clear advantage over multiple existing UAP defense methods relying on generating UAPs (e.g., (Akhtar et al., 2018; Mummadi et al., 2019; Borkar et al., 2020) etc.), for which, perturbations are generated for each target class. For datasets that contains a large number of classes (e.g., ImageNet dataset contains 1000 classes, JFT-300M dataset (Sun et al., 2017) contains 18k classes), a large number of perturbations shall be generated in order to achieve acceptable defense performance. Unlike these methods, *Democratic Training* transforms some clean samples into low-entropy samples and and it does not require the size of clean set to be large ($\leq 5\%$ of training set).

## 4 IMPLEMENTATION AND EVALUATION

In the following, we conduct multiple experiments to evaluate the effectiveness of *Democratic Training* by answering multiple research questions (RQs). All experiments are conducted on a machine with 96-Core 1.4GHz CPU and 60GB system memory with an NVIDIA 24GB RTX 4090 GPU. Our approach has been implemented as a self-contained toolkit in Python and is open-sourced (https://gitlab.com/sunbing7/democratic_training).

Table 1: Neutral Networks Used.

| Net | Dataset | Architecture | Acc |
|-----|---------|--------------|-----|
| $NN_1$ | ImageNet | ResNet50 | 0.73 |
| $NN_2$ | ImageNet | VGG19 | 0.70 |
| $NN_3$ | ImageNet | GoogleNet | 0.69 |
| $NN_4$ | ASL | MobileNet | 0.99 |
| $NN_5$ | CalTech101 | ShuffleNetV2 | 0.85 |
| $NN_6$ | EuroSAT | ResNet50 | 0.89 |
| $NN_7$ | CIFAR-10 | WideResNet | 0.93 |

Table 2: UAP Defense Performance.

| Model | Before | | After | | | |
|-------|--------|----|-------|----|--------------|------|
| | AAcc. | SR | AAcc. | SR | $\Delta CAcc.$ | Time |
| $NN_1$ | 0.134 | 0.714 | 0.617 | 0.002 | -0.02 | 8 |
| $NN_2$ | 0.067 | 0.701 | 0.431 | 0.077 | -0.04 | 36 |
| $NN_3$ | 0.195 | 0.584 | 0.549 | 0.004 | -0.03 | 7 |
| $NN_4$ | 0.035 | 0.997 | 0.894 | 0.004 | -0.02 | 18 |
| $NN_5$ | 0.059 | 0.842 | 0.715 | 0.018 | -0.01 | 30 |
| $NN_6$ | 0.236 | 0.784 | 0.786 | 0.048 | -0.01 | 16 |
| $NN_7$ | 0.154 | 0.933 | 0.860 | 0.031 | -0.03 | 21 |
| **Avg** | **0.126** | **0.794** | **0.693** | **0.028** | **-0.02** | **19** |

We report attack success rate (SR), adversarial accuracy (AAcc.), change in accuracy on clean inputs ($\Delta CAcc.$) and execution time (Time) in minutes.

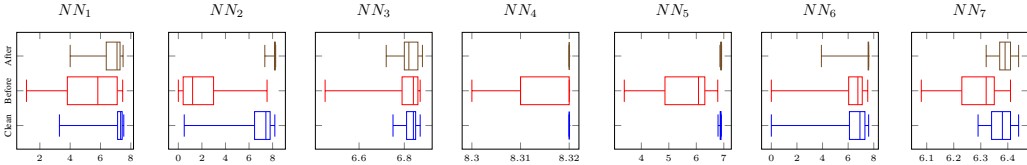

Figure 3: Change in layer-wise entropy.

## 4.1 EXPERIMENT SETUP

We conduct our experiments with 7 neural network models trained over 5 benchmark datasets: 1) *ImageNet (Deng et al., 2009)*, 2) *ASL Alphabet (Sau, 2018)*, 3) *Caltech101 (Li et al., 2022)*, 4) *EuroSAT (Helber et al., 2019)* and 5) *CIFAR-10 (Krizhevsky, 2009)*. Details can be found in Appendix 8.3. For experiments with the ImageNet dataset, we adopt the pretrained models from PyTorch (Paszke et al., 2019). For experiments with ASL, Caltech101, EuroSAT and CIFAR-10 datasets, we train CNN models following standard model training process. Details of the models are summarized in Table 1. When applying *Democratic Training*, we focus on the last pooling or dense layer for the entropy calculation since the effect of UAP on layer-wise entropy is stronger in deep layers as shown in Section 2. A small set of clean data ($\leq 5\%$ of the training set) is used during the model enhancement.

## 4.2 RESEARCH QUESTIONS AND ANSWERS

***RQ1**: Is* Democratic Training *effective in defending against UAP attacks?*

For each neural network, we train eight UAPs for randomly selected targets. The details of the UAP attacks are summarized in Table 2. We systematically apply *Democratic Training* to all the above-mentioned models and return the repaired models $NN_1'$ to $NN_7'$.

Firstly, we measure the layer-wise entropy of clean and UAP perturbed inputs on the repaired models. Figure 3 shows the box plot for the layer-wise entropy of clean samples and UAP infected samples before and after applying *Democratic Training*. On average, across all original models and attack target classes, the layer-wise entropy difference between clean inputs and those perturbed by UAPs is $16.7\%$. After applying *Democratic Training*, such difference is reduced to $0.2\%$. Thus, *Democratic Training* is able to reduce the effect of UAPs in terms of layer-wise entropy effectively.

Next, we show the change in UAP attack success rate (SR) and model accuracy (on clean inputs (Clean Acc.) and perturbed inputs (AAcc.)). As shown in Table 2, on average, across all original models and attack target classes, the attack success rate is reduced from $79.4\%$ to $2.8\%$ after applying *Democratic Training*. In addition, the adversarial accuracy is improved from $12.6\%$ to $69.3\%$. Hence, by reducing the effect of UAPs on layer-wise entropy, the effectiveness of UAPs is reduced significantly. In terms of clean sample accuracy, it is minimally affected. On average, the model accuracy is reduced by about $2\%$. Thus, *Democratic Training* is able to focus on removing the effect of UAPs while the model functionality is maintained.

Table 3: Performance of *Democratic Training* on UAPs generated using sPGD, LaVAN, GAP and SGA.

| Model | sPGD Before | | sPGD After | | LaVAN Before | | LaVAN After | | GAP Before | | GAP After | | SGA Before | | SGA After | |
|---|---|---|---|---|---|---|---|---|---|---|---|---|---|---|---|---|
| | AAcc. | SR | AAcc. | SR | AAcc. | SR | AAcc. | SR | AAcc. | SR | AAcc. | SR | AAcc. | SR | AAcc. | SR |
| $NN_1$ | 0.231 | 0.594 | 0.629 | 0.007 | 0.152 | 0.790 | 0.677 | 0.005 | 0.229 | 0.512 | 0.575 | 0.002 | 0.133 | 0.722 | 0.592 | 0.004 |
| $NN_2$ | 0.248 | 0.484 | 0.552 | 0.016 | 0.058 | 0.506 | 0.343 | 0.041 | 0.144 | 0.393 | 0.482 | 0.001 | 0.067 | 0.806 | 0.415 | 0.096 |
| $NN_3$ | 0.388 | 0.281 | 0.592 | 0.000 | 0.086 | 0.848 | 0.621 | 0.002 | 0.226 | 0.460 | 0.614 | 0.000 | 0.147 | 0.640 | 0.510 | 0.010 |
| $NN_4$ | 0.045 | 0.980 | 0.981 | 0.000 | 0.537 | 0.271 | 0.678 | 0.033 | 0.031 | 0.921 | 0.984 | 0.034 | 0.034 | 0.999 | 0.904 | 0.009 |
| $NN_5$ | 0.326 | 0.270 | 0.790 | 0.008 | 0.337 | 0.449 | 0.867 | 0.001 | 0.128 | 0.751 | 0.725 | 0.006 | 0.106 | 0.797 | 0.743 | 0.020 |
| $NN_6$ | 0.401 | 0.375 | 0.861 | 0.022 | 0.224 | 0.743 | 0.925 | 0.004 | 0.274 | 0.757 | 0.900 | 0.106 | 0.227 | 0.776 | 0.811 | 0.033 |
| **Avg** | **0.273** | **0.497** | **0.734** | **0.009** | **0.232** | **0.601** | **0.685** | **0.014** | **0.172** | **0.632** | **0.713** | **0.025** | **0.119** | **0.790** | **0.662** | **0.029** |

Hence, to answer RQ1, *Democratic Training* is able to reduce the attack success rate of UAP attacks and improve the robustness against adversarial samples effectively, and at the same time, the model accuracy is maintained at a high level.

***RQ2**: Is* Democratic Training *effective in mitigating UAPs crafted from different attack methods?*

There are many UAP generation algorithms proposed in existing works and we further evaluate *Democratic Training* against another four types of UAP attacks: 1) sPGD (Mummadi et al., 2019) which adopts PGD to update the perturbations iteratively to generate UAPs, 2) LaVAN (Karmon et al., 2018) which is proposed as a method to generate image-agnostic localized adversarial noise that covers only $2\%$ of the image but fool the neural network, 3) GAP (Poursaeed et al., 2018) as a generator-based UAP attack method which adopts generative model for crafting UAPs and 4) SGA (Liu et al., 2023) which alleviates the gradient vanishing and escapes from poor local optima when generating UAP. For each method, we randomly select eight attack target classes and train UAPs for $NN_1$ to $NN_6$ and evaluate their attack success rate and model accuracy before and after applying *Democratic Training*. The average results across all models and target classes are summarized in Table 3. For all models, on average, sPGD attack achieves $49.7\%$ targeted success rate and the adversarial accuracy is below $27.3\%$. LaVAN attack achieves $60.1\%$ success rate with adversarial accuracy of $23.2\%$. GAP attack achieves $63.2\%$ success rate and the adversarial accuracy is below $17.2\%$. SGA attack achieves $79.0\%$ success rate and the adversarial accuracy is below $11.9\%$. When tested on *Democratic Training* enhanced models, the adversarial accuracy is improved to $73.4\%$, $68.5\%$, $71.3\%$ and $66.2\%$ for sPGD, LaVAN, GAP and SGA attacks respectively, and the attack success rate is below $0.9\%$, $1.4\%$, $2.5\%$ and $2.9\%$. Thus, models enhanced by *Democratic Training* are robust against UAPs generated in different ways. We further analyze the change in layer-wise entropy before and after applying *Democratic Training*. The results are summarized in Appendix 8.6. These results suggest that, regardless of the generation method, the effect of targeted UAP on a model can be revealed by layer-wise entropy and such effect can be suppressed via our entropy based model enhancement effectively.

Thus, to answer RQ2, *Democratic Training* is effective at defending UAPs generated with various algorithms.

***RQ3**: How does* Democratic Training *compare with adversarial training?*

Adversarial training can be a useful method to improve model robustness against UAPs (Mummadi et al., 2019; Shafahi et al., 2020; Benz et al., 2021). We evaluate the effectiveness of low-entropy samples and adversarial samples in finetuning a given model three settings: 1) non-targeted adversarial training, i.e., adversarial examples are not targeted and are generated on-the-fly , 2) targeted adversarial training, i.e., adversarial examples are targeted and are generated on-the-fly and 3) finetuning with pretrained targeted UAP. While there are various options of adversarial training algorithm for the first two settings, we adopt PGD based adversarial training (Madry et al., 2018) as it provides a good trade-off between being computationally efficient and powerful (Mummadi et al., 2019). We finetune the model with adversarial samples generated with the same number of iterations as in *Democratic Training* for a fair comparison. Furthermore, we assume the attack target class is known for targeted-adversarial finetuning (which gives the defender some unrealistic advantage). For finetuning with pretrained UAPs, similarly we assume the target class is known and train a set of 10 UAPs to be used together with a set of clean samples. During the finetuning, we add a randomly chosen pretrained UAP to a clean sample with $50\%$ probability. We keep the number of clean examples used in finetuning the same as *Democratic Training* as well. The average performance over all

Table 4: Performance of adversarial training.

| Setting | AAcc. | SR | $\Delta$CAcc. |
|---|---|---|---|
| Targeted | 0.464 | 0.167 | -0.104 |
| Non-targeted | 0.295 | 0.455 | -0.168 |
| Known UAP | 0.476 | 0.223 | 0.0 |
| TRADES | 0.816 | 0.022 | -0.110 |

Table 5: Performance of existing methods.

| Method | AAcc. | SR | $\Delta$CAcc. |
|---|---|---|---|
| SFR | 0.468 | 0.011 | -0.022 |
| CFN | 0.150 | 0.559 | -0.073 |
| FNS | 0.149 | 0.623 | -0.013 |
| DensePure | 0.802 | 0.010 | -0.121 |

6 models is shown in Table 4. All three methods are not that effective in UAP defense, i.e., the attack success rate remains high ($> 16\%$) and adversarial accuracy is lower than $50\%$ after the finetuning. In comparison, *Democratic Training* is able to reduce the attack success rate to $< 3\%$ and improve the adversarial accuracy to $69\%$ on average. We believe this is due to the fact that adversarial training aims to direct adversarial examples towards their correct predictions while *Democratic Training* focuses on guiding low-entropy samples. Based on our experimental results, low-entropy samples are more efficient in guiding the model enhancement process.

Moreover, we evaluate the performance of a well-recognized adversarial training method TRADES (Zhang et al., 2019) on UAP defense. As shown in the last row of Table 4. TRADES is effective in defending against UAPs but sacrifices model accuracy for over $10\%$.

Hence, to answer this RQ3, *Democratic Training* is more effective in defending against UAPs when compared to adversarial training with equivalent parameter settings.

***RQ4***: *How does* Democratic Training *compare with other existing neural network UAP defense methods?*

We further compare the performance of *Democratic Training* with four state-of-the-art UAP defense methods, i.e., selective feature regeneration (SFR) (Borkar et al., 2020), clipping feature norms (CFN) (Yu et al., 2021), feature norm suppressing (FNS) (Yu et al., 2023) and DensePure (Xiao et al., 2023). SFR is an approach proposed to defend against UAPs from feature-level. It deploys feature regeneration units in a given model aiming to transform vulnerable features into resilient features against UAPs. CFN is proposed based on the fact that universal adversarial patches usually lead to deep feature vectors with very large norms (Yu et al., 2021). It introduces a feature norm clipping layer to be inserted into the original model that aims to adaptively suppress the generation of large norm deep feature vectors. Similarly, FNS is designed on top of CFN which is able to renormalize the feature norm by non-increasing functions. FNS can be adaptively inserted in to a given model to achieve multistage suppression of the generation of large norm feature vectors. No training is required for such feature norm suppressing layer. DensePure employs iterative denoising to an input image to get multiple reversed samples with different random seeds. Next, the samples are given to the model to make final decision via majority voting. The results of the average performance for SFR, CFN, FNS and DensePure are summarized in Table 5. Both CFN and FNS are not effective in defending against UAPs, i.e., the attack success rates remain above $50\%$. Thus, suppressing or clipping feature norms of a given model has limited effect on weakening the impact of targeted UAPs. Furthermore, both CFN and FNS modify the original neural network architecture by inserting an additional feature norm clipping / suppression layer. Although SFR achieves comparable UAP defense performance as *Democratic Training*, it modifies the architecture of the original model which is often not preferred in real-life application (as this might prolong development cycles and bring in integration challenges (Hutter et al., 2019)). Furthermore, SFR requires to pretrain 25 UAPs (and 2000 synthetic UAPs) to train the additional layers, which is rather time consuming (it takes $> 40$ min to train one UAP based on method proposed in (Moosavi-Dezfooli et al., 2017) following open-source implementation[1] while *Democratic Training* repairs the same model within 10 min). DensePure is effective in improving the model robustness against UAPs but model accuracy is affected which drops by $12.1\%$. Moreover, it introduces overhead in inference time for reversed samples.

Hence, to answer RQ4, *Democratic Training* is more effective in mitigating the impact of UAPs on trained neural networks, which does not require to change the original model architecture.

---

[1]https://github.com/qilong-zhang/Pytorch_Universal-adversarial-perturbation

## 5 RELATED WORKS

**Adversarial attacks.** Neural networks are highly vulnerable to adversarial attacks, which are small, deliberately crafted perturbations to input data that can fool the model into making incorrect predictions. Such perturbation can be 1) image-specific where the attacker computes a perturbation for every clean input and 2) image-agnostic where a single perturbation will cause majority of clean samples to fool a given model. In recent years, many input-specific adversarial attacks are proposed to generate disruptive perturbations. Goodfellow et al. (2015) introduce Fast Gradient Sign Method (FGSM) that generates adversarial examples. Subsequently, Basic Iterative Method (BIM) is proposed (Kurakin et al., 2017) as an extension of FGSM which applies small FGSM steps iteratively aiming to generate higher quality perturbations. Madry et al. (2018) propose PGD which optimizes the perturbation at each iteration based on gradient of loss function. Furthermore, Carlini & Wagner (2017) propose C&W attack which formulates the adversarial example generation as an optimization problem aiming at minimal perturbation. Together with many others, e.g., (Wang et al., 2021a; Wu et al., 2020; Zhong & Deng, 2021; Dong et al., 2019; Wang et al., 2022; Peng et al., 2021), adversarial attacks pose a significant threat to real-world applications in different domains.

**Universal adversarial attacks.** Unlike per-instance perturbations, UAPs work for the majority of clean samples, i.e., adding a single perturbation to majority of clean samples, the neural network will response with incorrect predictions. Such attacks can be broadly classified in to noise-based and generator-based attacks. Noise-based attack methods directly train a UAP that can be applied to all inputs while generator-based methods train an extra generative model as a bridge to craft he perturbation indirectly (Weng et al., 2024). Moosavi-Dezfooli et al. (2017) first explore the existence of such input-agnostic adversarial perturbations. Furthermore, Khrulkov & Oseledets (2018) propose to craft UAPs by maximizing the difference between the activations of a hidden layer for clean and perturbed inputs. Later on, many noise-based methods are proposed with good performance (Mopuri et al., 2017; Zhang et al., 2020b). On the other hand, Poursaeed et al. (2018) firstly apply generative model for crafting UAPs. NAG is proposed (Mopuri et al., 2018) with a novel loss function for training the perturbation generator. Beyond above mentioned methods, there are many other UAP attacks, e.g., (Benz et al., 2020; Khrulkov & Oseledets, 2018; Zhang et al., 2020a; Amada et al., 2021; Sun et al., 2024). Compared to input-specific perturbations, UAPs are more efficient in terms of computation cost and become a more significant threat in practice.

**Defense against adversarial attacks.** Defense against adversarial attacks can be grouped into six domains (Costa et al., 2024): 1) adversarial training which augments the training data with adversarial examples to make the model more robust (Goodfellow et al., 2015; Madry et al., 2018; Wong et al., 2020; Zhang et al., 2019; Mummadi et al., 2019; Chen et al., 2022), 2) modifying the training process which adjusts the training process to improve robustness (Papernot et al., 2016; Shafahi et al., 2020; Huang et al., 2020; Chen & Lee, 2021; Pang et al., 2022; Akhtar et al., 2018), 3) use of supplementary networks which add extra networks on top of the original model to remove the effect of adversarial perturbations (Liu et al., 2020; Liao et al., 2018; Li et al., 2021; Abusnaina et al., 2021; Ho & Vasconcelos, 2022; Borkar et al., 2020), 4) changing network architecture which modifies the architecture of the original model for robustness (Xie et al., 2019; Guo et al., 2020; Xie et al., 2018; Atzmon et al., 2019; Yu et al., 2021; 2023), 5) performing network validation which validates and certifies the robustness of a given model (Pei et al., 2019; Ma et al., 2018; Kim et al., 2019) and 6) adversarial purification which removes adversarial perturbations of input samples and recovers the clean image (Gowal et al., 2021; Ho et al., 2020; Sehwag et al., 2022; Nie et al., 2022; Xiao et al., 2023). Among them, there are multiple works proposed to defense against UAPs (Moosavi-Dezfooli et al., 2017; Akhtar et al., 2018; Mummadi et al., 2019; Borkar et al., 2020; Yu et al., 2021; 2023).

## 6 CONCLUSION

In conclusion, we propose *Democratic Training* as an efficient and effective defense method against targeted UAP attacks for neural networks. *Democratic Training* first analyzes the layer-wise entropy to understand how UAP deceive the model and conducts entropy-based model enhancement to mitigate the effect of UAP. Our experimental results show that *Democratic Training* is effective in removing the effects of UAPs from a given model and it outperforms existing state-of-the-art UAP attack defense methods.

## 7    ACKNOWLEDGEMENTS

This research is supported by Singapore Ministry of Education under its Academic Research Fund Tier 3 (Award ID: MOET32020 − 0004).

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

# 8 APPENDIX

## 8.1 FUTURE WORKS

In our future works, we would like to extend *Democratic Training* to other model architectures (e.g., transformer based models) and non-vision tasks (e.g., language models, audio tasks). Moreover, we would like to integrate *Democratic Training* with adversarial training, i.e., apply low-entropy samples in adversarial training. We would like to explore if the performance can be further improved and whether the method can be extended to other types of adversarial attacks.

## 8.2 MEASURING ENTROPY IN NEURAL NETWORKS

Barbiero et al. (2022) proposed to apply an entropy-based layer to conduct logic explanations of neural networks. For a concept-based classifier where human-understandable input concepts are mapped to output predictions, the relevance of an input concept $j$ to a prediction class $i$ can be approximated by the weight connecting $j^{th}$ input to $i_{th}$ class embedding, i.e.,

$$\gamma_j^i = ||W_j^i||_1$$

$$\beta_j^i = \frac{e^{\frac{\gamma_j^i}{\tau}}}{\sum_l e^{\frac{\gamma_l^i}{\tau}}} \tag{7}$$

where $W$ represents the weight matrix and $\tau$ is a user-defined temperature parameter to tune the softmax function. The entropy of distribution $\beta^i$

$$H(\beta^i) = -\sum_j \beta_j^i \log \beta_j^i \tag{8}$$

is minimized when a single input concept dominates the prediction and it is maximized when all concepts are equally important.

Wan et al. (2019) proposed entropy-based pooling for CNNs that helps the network to concentrate on semantically important image regions. In CNN architecture, a global averaging pooling (GAP) layer is typically connected to a fully connected (FC) layer with softmax activation to produce the

class scores. The input to GAP layer is the last convolutional feature maps $U \in \mathbb{R}^{h \times w \times c}$ consisting of local feature vectors $v_i \in \mathbb{R}|i = 1, 2, \cdots, hw$, and the final prediction scores are computed as

$$f_{GAP}(U) = \frac{1}{hw} \sum_i v_i$$

$$F = W^T f_{GAP}(U) \tag{9}$$

$$= \frac{1}{hw} \sum_i W^T v_i$$

The entropy of the localized class probability for location $i$ is then measured by

$$p_i = softmax(W^T v_i)$$

$$H(p_i) = -\sum_k p_i(k) \log p_i(k) \tag{10}$$

where $W^T v_i \in \mathbb{R}^K$. For a feature location $i$, if its receptive field is centered on a specific object, the localized class prediction of $v_i$ should probably be highly confident leading to a low entropy value measured using Equation 10. Otherwise, if its receptive field is centered on image textures or patterns that frequently occurred in other image classes, the corresponding entropy should generally be high (Wan et al., 2019).

## 8.3 DATASETS USED IN OUR EXPERIMENTS

- *ImageNet (Deng et al., 2009):* The ImageNet 2012 dataset, also known as the ILSVRC 2012 (ImageNet Large Scale Visual Recognition Challenge), is a large-scale dataset used for visual object recognition tasks. It contains over 1.2 million images for training, 50,000 for validation, and 100,000 for testing. There are 1,000 different classes, which include a wide variety of objects, animals, and scenes. Each class has hundreds to thousands of images. We focus on image classification task in this work.

- *ASL Alphabet (Sau, 2018):* This dataset is a collection of images of alphabets from the American Sign Language. It consists of 87K $200 \times 200$ images of 29 classes, including 26 letters (A to Z) and 3 classes for "SPACE", "DELETE" and "NOTHING". The task is to identify the 29 alphabets.

- *Caltech101 (Li et al., 2022):* This dataset contains of 9k pictures of objects belonging to 101 categories. There are 40 to 800 images per category. Images are of variable sizes with typical edge lengths of 200 to 300 pixels. The task is to recognize the 101 different objects.

- *EuroSAT (Helber et al., 2019):* This dataset is a benchmark dataset in the field of remote sensing and geospatial analysis for the classification of land use and land cover from satellite imagery. It contains 27k $64 \times 64$ labeled images of 10 different classes representing various land use and land cover types, including: forest, highway, river etc. The task is to classify the land usage types.

- *CIFAR-10 (Krizhevsky, 2009):* This dataset is a widely used benchmark dataset for image classification in machine learning. It contains 60k color images, each with a resolution of 32x32 pixels. The task is for image recognition of 10 catagories.

## 8.4 PERFORMANCE ON UAPS GENERATED WITH DIFFERENT $\epsilon$.

To further evaluate *Democratic Training*, we generate UAPs with different $\epsilon$ settings. We train UAPs with $\epsilon = 5/255$ and eight with $\epsilon = 15/255$ for $NN_1$ with the same set of target classes selected in RQ1 and evaluates the attack success rate and model accuracy on the *Democratic Training* repaired model (repaired with $\epsilon = 10/255$). The average results are summarized in Table 6. For a smaller perturbation budget ($\epsilon = 5/255$) the repaired model stays robust against the generated UAPs. The attack success rate is below $1\%$ for all targeted classes. For a larger perturbation budget ($\epsilon$ is larger than the value used during the finetuning process) where $\epsilon = 15/255$, the repaired model still remains robust to a certain level. The average attack success rate drops from $91.3\%$ to $11.5\%$. For the adversarial examples to be human-imperceptible, the $\epsilon$ shall not be large. Hence, by setting it

Table 6: Performance of *Democratic Training* on UAPs generated with different $\epsilon$.

| $\epsilon$ | Before | | After | |
|---|---|---|---|---|
| | AAcc. | SR | AAcc. | SR |
| 5/255 | 0.468 | 0.274 | 0.699 | 0.000 |
| 10/255 | 0.134 | 0.714 | 0.617 | 0.002 |
| 15/255 | 0.027 | 0.913 | 0.364 | 0.115 |

Table 7: Adaptive attack performance.

| Model | AAcc. | SR |
|---|---|---|
| $NN_1'$ | 0.480 | 0.115 |
| $NN_2'$ | 0.344 | 0.288 |
| $NN_3'$ | 0.491 | 0.058 |
| $NN_4'$ | 0.655 | 0.174 |
| $NN_5'$ | 0.385 | 0.409 |
| $NN_6'$ | 0.559 | 0.369 |

to a reasonable value, e.g., $\epsilon = 10$, the *Democratic Training* repaired model will be robust against UAPs generated with various perturbation budgets.

Thus, *Democratic Training* repaired model stays robust against UAP attacked samples generated with different perturbation budgets ($\epsilon$).

## 8.5 ADAPTIVE ATTACKS

In this section, we evaluate *Democratic Training* on two types of adaptive UAP attacks: 1) secondary white-box attacks, where the attacker has full access to the *Democratic Training* repaired model, and 2) advanced attacks where the attacker is capable of tailoring the UAP trying to bypass our defense.

Firstly, for secondary attacks, for pretrained model $NN_1$ to $NN_6$ described in Table 1, we apply *Democratic Training* to repair it as in RQ1 to mitigate the effect of UAPs and obtain the repaired models. Next, we apply the method DF-UAP proposed in (Zhang et al., 2020b) on all the repaired models ($NN_1'$, $NN_2'$, $NN_3'$, $NN_4'$, $NN_5'$ and $NN_6'$.) to generate new sets of UAPs accordingly. We keep all the attack parameters the same as the initial attack including the attack target classes. The secondary attack performance is show in Table 7. As described in Section 4, before applying *Democratic Training*, the UAP attack (Zhang et al., 2020b) can easily achieve an average of $81.3\%$ targeted attack success rates and $14.7\%$ adversarial accuracy. After applying *Democratic Training*, a subsequent attack can achieve an average attack success rate of $23.6\%$ (with highest attack success rate of $40.9\%$ on $NN_5'$ and lowest of $5.8\%$ on $NN_3'$). The average adversarial accuracy on the subsequent attack is $48.6\%$. Furthermore, we apply sPGD and GAP attacks on $NN_1'$ as well. Adaptive sPGD is able to achieve $16.5\%$ attack success rate and $48.7\%$ adversarial accuracy. Adaptive GAP only manages to achieve an average success rate of $1.1\%$ and the adversarial accuracy stays above $50\%$. Hence, similar to DF-UAP, subsequent UAP attacks such as sPGD and GAP are no longer effective on *Democratic Training* repaired models. Based on such result, we believe that, as UAPs exploit large correlations and redundancies in the decision boundary of a given model (Moosavi-Dezfooli et al., 2017), *Democratic Training* is able to reduce such correlations and redundancies so that it is much more difficult to find highly effective UAPs on the *Democratic Training* enhanced models.

Thus, although secondary UAP attacks on *Democratic Training* repaired models can still generate UAPs that successfully fool the models, our defense keeps the secondary attack success rate to a very low level while keeping the adversarial accuracy high. Hence, based on above results, *Democratic Training* repaired model is able to stay robust against adaptive UAP attacks.

Secondly, for advanced attacks we conduct experiments such that when generating UAPs, the attacker further controls the change in layer-wise entropy. Based on DF-UAP, the optimization loss function used for crafting an UAP is modified as:

$$\mathbf{L}(i) = (1 - \rho) \cdot \mathbf{L}_{cce}(i, y_t) - \rho \cdot \mathbf{H}(i) \qquad (11)$$

where $i$ represents a training sample, $y_t$ represents the attack target class and $\mathbf{H}(i)$ represents the layer-wise entropy loss for $i$. We use $\mathbf{H}(i)$ to control the entropy change caused by the UAP and parameter $\rho$ is used to control the importance of $\mathbf{H}(i)$ over attack success rate. We conduct such advanced attack on model $NN_1$ to $NN_6$ with $\rho$ set to 0.1 to 0.9. All models show similar results and for illustration purpose, results on $NN_1$ are summarized Table 8. Increasing $\rho$ causes the attack performance to drop, i.e., the attack success rate starts to drop when $\rho > 0.5$ and the attack SR is below $60\%$ when $\rho = 0.9$. Our defense stays effective across different $\rho$ settings where the attack SR is reduced to $< 1\%$ for all scenarios. Hence, knowing how *Democratic Training* enhance the

Table 8: Advanced attack performance on $NN_1$. We report the Adversarial accuracy (AAcc.), attack success rate (SR) and layer-wise entropy (Entropy) of UAP infected samples. Note that clean sample layer-wise entropy is 7.1

| $\rho$ | Before | | | After | | |
|---|---|---|---|---|---|---|
| | AAcc. | SR | Entropy | AAcc. | SR | Entropy |
| 0.0 | 0.118 | 0.764 | 5.62 | 0.619 | 0.001 | 7.39 |
| 0.1 | 0.121 | 0.775 | 6.07 | 0.619 | 0.001 | 7.39 |
| 0.2 | 0.118 | 0.759 | 6.29 | 0.619 | 0.001 | 7.40 |
| 0.3 | 0.128 | 0.764 | 6.35 | 0.613 | 0.001 | 7.39 |
| 0.4 | 0.127 | 0.761 | 6.89 | 0.612 | 0.001 | 7.40 |
| 0.5 | 0.125 | 0.759 | 7.07 | 0.608 | 0.0 | 7.39 |
| 0.6 | 0.141 | 0.745 | 7.13 | 0.618 | 0.0 | 7.39 |
| 0.7 | 0.161 | 0.693 | 7.16 | 0.599 | 0.001 | 7.39 |
| 0.8 | 0.185 | 0.657 | 7.33 | 0.609 | 0.001 | 7.40 |
| 0.9 | 0.207 | 0.568 | 7.43 | 0.628 | 0.0 | 7.39 |

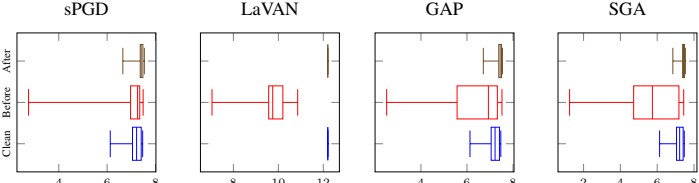

Figure 4: Change in layer-wise entropy of other UAPs.

model and control the change in layer-wise entropy during attack process, the adversary is still not able to bypass our defense effectively.

## 8.6    ENTROPY ANALYSIS ON OTHER UAPS

As part of RQ2, we further analyze the change in layer-wise entropy of clean and UAP infected samples for other types of UAP attacks, i.e., sPGD, LaVAN, GAP and SGA. The results are summarized in Figure 4, which show that similar to DF-UAP, UAPs generated with above mentioned four methods also cause the layer-wise entropy to drop and *Democratic Training* is able to mitigate such effect effectively.

## 8.7    NON-TARGETED UAP ATTACKS

We further evaluate *Democratic Training* on non-targeted UAP attacks. We generate non-targeted UAPs following DF-UAP for $NN_1$ to $NN_6$ and report the adversarial accuracy and attack success rate on original models ($NN_1$ to $NN_6$) and repaired models ($NN_1'$ to $NN_6'$). For non-targeted attacks, the attack success rate (SR) is calculated as $SR = \sum_{x \in X} \frac{|N(x+\delta) \neq N(x)|}{|X|}$, where $x \in X$ represents a clean sample, $\delta$ is the UAP. The results are summarized in Table 9.

Although not designed for non-targeted UAPs, *Democratic Training* manages to reduce the attack SR from over $90\%$ to $30\%$ on average. This is indeed not as effective as targeted UAP defense and we believe this is due to the different entropy spectrum caused by the two types of UAPs. Figure 5 shows the entropy spectrum of clean and non-targeted UAP infected samples for $NN_2$ where no clear separation of the two is observed.

Table 9: Performance on non-targeted UAP attacks.

| Model | Before | | After | | |
|-------|--------|------|-------|------|---------|
|       | AAcc.  | SR   | AAcc. | SR   | $\Delta CAcc.$ |
| $NN_1$ | 0.057 | 0.939 | 0.594 | 0.267 | -0.047 |
| $NN_2$ | 0.056 | 0.943 | 0.369 | 0.559 | -0.066 |
| $NN_3$ | 0.098 | 0.888 | 0.469 | 0.408 | -0.035 |
| $NN_4$ | 0.002 | 0.981 | 0.918 | 0.066 | -0.031 |
| $NN_5$ | 0.053 | 0.958 | 0.607 | 0.374 | -0.019 |
| $NN_6$ | 0.289 | 0.737 | 0.801 | 0.129 | -0.008 |
| **Avg** | **0.092** | **0.907** | **0.626** | **0.300** | **-0.034** |

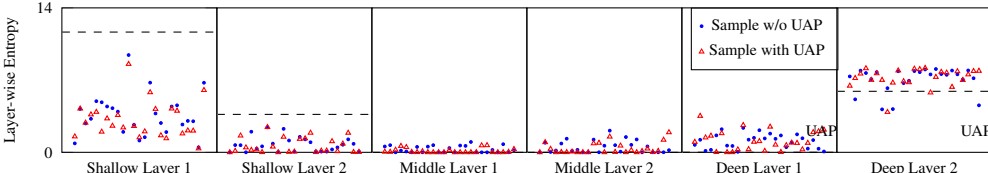

Figure 5: Layer-wise entropy of $NN_2$.

