# OpenReview forum: "Democratic Training Against Universal Adversarial Perturbations"
_ICLR.cc/2025/Conference — ICLR 2025 Poster_

### Official Review · Reviewer_etKh · 2024-10-18

**Soundness:** 2
**Presentation:** 2
**Contribution:** 2
**Rating:** 6
**Confidence:** 3

**Summary:**

The author aims to improve the robustness of universarial adversarial by fine-tuning with a small amount of data, mainly by performing entropy based data augmentation to suppress the influence of general adversarial perturbations in a given model. Then the experiment was presented.

**Strengths:**

1, The setting is reasonable, author want to resist universal adversarial samples through small cost, this may be useful in some situation.
2, Good experiment results.

**Weaknesses:**

1, The method provided in the article is not novel, overall, it is still based on the 'min-max' methods. And in my opinion, it seems to be a weakened version of adversarial training.

2, The symbols are somewhat confused. For example, in euqation (4), author use $L_{cce}$, I think this means cross entropy loss, but author did not introduce what is $L_{cce}$ before the euqation (4); in algorithm line 3, author define $I_b^{en}$ which is got by SampleGenerator, but do not use it in the following of algorithm, or maybe it have been wirtten as $i_{en}$ in line 4?

3, I try to test the author's algorithm on CIFAR10(VGG16, budget 8/255), but I didn't get such a good result shown in table2, the SR has only decreased by about 10%. Author did not submit the code, so I hope the author can provide a detailed introduction to the settings for Algorithms 1 and 2 (For example, how to select the epoch, learning rate, hyperparameters), and it's best to provide a detailed explanation of each symbol and steps in the algorithm.

**Questions:**

1: The author obtained the idea of Democratic Training by studying the performance of UAP between different layers in network. I want to know why author said 'Democratic Training does not rely on generating UAPs and are thus not limited to specific UAP attacks.', As mentioned in the article, the UAP analyzed by the author is produced based on FA-UAP, so the properties analyzed should mainly for such kind of UAP, and the algorithm guided by this should also mainly target on FA-UAP. If it cannot be argued that all UAPs have such  properties, then this statement seems unreasonable?

2: The author did not provide a detailed explanation for $H(i)$ in algorithm 2 line 2, according to the equation (3), I think that the author is  trying to say $H(i)$ is an entropy loss. Less strictly speaking, maximizing H(x) means that the various components of x are directly averaged as much as possible. So, is the goal of SG(SampleGenerator, Algorithm 2) equivalent to finding an adversarial sample with each component average? Is this a form of weakened PGD? If so, why is finding weaker adversarial samples beneficial for improving robustness? Why not directly target finding UAP for SG?

---

> ### Author Response · Authors · 2024-11-22
> **Response to Reviewer etKh I**
>
> We appreciate your thoughtful review and the constructive feedback provided. Below are our responses to the points you've raised.
>
> Response to weaknesses:
> 1. Although it is right to say that our method is based on min-max, we would respectfully argue that our observation and method are still novel - as it is based on entry spectrum rather than adversarial attacks. That is, based on our observation that UAP causes abnormal entropy, Democratic Training mitigates the effect of UAPs in general by finetuning a given model with low entropy samples generated on-the-fly. Different from adversarial training which trains a model from scratch, Democratic Training improves the robustness of a given model against UAP attacks more efficiently. Based on our experimental results in RQ3, our low entropy samples are more effective compared with adversarial examples when finetune a given model against UAP attacks.
> To further compare Democratic Training and adversarial training, we evaluate TRADES[1] which is a widely recognized adversarial training method on UAP defense and the results are shown below:
>
> | | before || after (ours) || after (TRADES) ||
> | ------ | ----- | ----- | ----- | ----- | ----- | ----- |
> | target | Aacc. | SR    | Aacc. | SR    | Aacc. | SR    |
> | 0      | 0.155 | 0.922 | 0.871 | 0.009 | 0.819 | 0.018 |
> | 1      | 0.144 | 0.927 | 0.864 | 0.016 | 0.816 | 0.004 |
> | 2      | 0.143 | 0.954 | 0.857 | 0.050 | 0.816 | 0.028 |
> | 3      | 0.116 | 0.972 | 0.875 | 0.046 | 0.819 | 0.042 |
> | 4      | 0.148 | 0.938 | 0.874 | 0.027 | 0.818 | 0.015 |
> | 5      | 0.226 | 0.861 | 0.861 | 0.036 | 0.818 | 0.042 |
> | 6      | 0.139 | 0.963 | 0.860 | 0.048 | 0.811 | 0.036 |
> | 7      | 0.143 | 0.942 | 0.849 | 0.045 | 0.812 | 0.010 |
> | 8      | 0.171 | 0.928 | 0.852 | 0.047 | 0.815 | 0.015 |
> | 9      | 0.134 | 0.954 | 0.847 | 0.014 | 0.817 | 0.010 |
> | avg    | 0.152 | 0.936 | 0.861 | 0.034 | 0.816 | 0.022 |
> | | Clean Acc: 0.931 || Clean Acc: 0.901 || Clean Acc: 0.827 ||
>
> Based on the above result, both TRADES and Democratic Training are effective in mitigating the effect of UAPs. However, TRADES sacrifices model accuracy for over 10% while Democratic Training keeps the model accuracy high (reduced by 3%). Furthermore, Democratic Training repairs the model within 20min while it takes more than 15hrs for TRADES to train a robust model on the same machine.
> Hence, adversarial training is effective in keeping models robust against UAP attacks but not time efficient. Democratic training finetunes a given model for a few epochs which is much faster and is able to reduce the attack success rate effectively while keeping the model accuracy on clean samples at a high level
>
> 2. Yes, $L_cce$ represents cross entropy loss and $I_b^{en}$ represents a batch of generated low entropy samples. Thanks for pointing this out and we will improve our presentation in the revised version.
>
> 3. Thanks for the suggestion and the effort in trying out our approach. We tested our approach on a wideresnet model training on CIFAR10 and below table shows the result. We are glad and preparing to open source our code (https://anonymous.4open.science/r/democratic_training-EB5A/ ).
>
> | | before || after ||
> | ------ | ----- | ----- | ----- | ----- |
> | target | Aacc. | SR    | Aacc. | SR    |
> | 0      | 0.155 | 0.922 | 0.871 | 0.009 |
> | 1      | 0.144 | 0.927 | 0.864 | 0.016 |
> | 2      | 0.143 | 0.954 | 0.857 | 0.050 |
> | 3      | 0.116 | 0.972 | 0.875 | 0.046 |
> | 4      | 0.148 | 0.938 | 0.874 | 0.027 |
> | 5      | 0.226 | 0.861 | 0.861 | 0.036 |
> | 6      | 0.139 | 0.963 | 0.860 | 0.048 |
> | 7      | 0.143 | 0.942 | 0.849 | 0.045 |
> | 8      | 0.171 | 0.928 | 0.852 | 0.047 |
> | 9      | 0.134 | 0.954 | 0.847 | 0.014 |
> | avg    | 0.152 | 0.936 | 0.861 | 0.034 |
> | Clean Acc. | 0.931 || 0.901 ||

---

> ### Author Response · Authors · 2024-11-22
> **Response to Reviewer etKh II**
>
> Response to questions:
> 1. Thanks for the comments. Indeed, our initial observation was based on DF-UAP (as described in Section 3.2). However, after we evaluated our approach against multiple UAP attack methods besides DF-UAP, i.e., sPGD, LaVAN and GAP, it becomes apparent that our method indeed works for all the UAP attacks that we tested. In fact, we further evaluated one recent UAP attack SGA [4] and the result of SGA is shown below:
>
> | | before || after  ||
> | ------- | ----- | ----- | ----- | ----- |
> | model   | Aacc  | SR    | Aacc  | SR    |
> | NN1     | 0.133 | 0.722 | 0.592 | 0.005 |
> | NN2     | 0.067 | 0.806 | 0.415 | 0.096 |
> | NN3     | 0.147 | 0.641 | 0.510 | 0.011 |
> | NN4     | 0.034 | 0.999 | 0.904 | 0.009 |
> | NN5     | 0.107 | 0.798 | 0.743 | 0.020 |
> | NN6     | 0.227 | 0.776 | 0.812 | 0.034 |
> | average | 0.119 | 0.790 | 0.663 | 0.029 |
>
> Moreover, we analyse the layer-wise entropy on clean samples and samples with UAP on original model and defended model. Comparing the entropy spectrum of clean samples and UAP perturbed samples on the original model, sPGD, LaVAN, GAP and SGA will cause layer-wise entropy to drop. While for UAP perturbed samples on Democratic Training enhanced model looks similar to that of clean samples, which suggest that our defense is able to mitigate the effect of different types of UAPs on entropy spectrum. We will include the entropy plots in the revised version.
>
> 2. Thanks for the comments, $H(i)$ represents the layer-wise entropy described in equation (3) for sample $i$. We will improve our presentation in the revised version. Different from generating a UAP, SG generates low entropy samples for model finetuning. Since we focus on targeted UAP attacks, methods relying on generated UAPs often require generating UAPs of a set of target classes when the attack target is unknown. However, SampleGenerator does not need target class information. Furthermore, we compare the effectiveness of adversarial examples and low entropy samples generated by SampleGenerator when finetuning a given model in RQ3. The results show that, with the same finetuning process setting, adversarial examples are less effective compared with low entropy samples in mitigating the effect of UAPs. Hence, based on above results, low entropy samples generated by our SampleGenerator are more powerful compared to adversarial examples and UAPs, which does not require any information on the attack target class.
>
> References
>
> [1] Costa, J. C., Roxo, T., Proença, H., & Inácio, P. R. (2024). How deep learning sees the world: A survey on adversarial attacks & defenses. IEEE Access.
>
> [4] Liu, Xuannan, et al. "Enhancing generalization of universal adversarial perturbation through gradient aggregation." Proceedings of the IEEE/CVF International Conference on Computer Vision. 2023.

---

> ### Comment · Reviewer_etKh · 2024-11-25
>
> Most of the conclusions in the author's paper and response come from experimental results, and I am willing to believe in the author's experimental results, so my problem has been solved and I will improve my score. But considering that I haven't seen the author's code and detailed experimental setup, my repeated experiments on other datasets are not satisfactory, so I have to lower my confidence.

---

> > ### Author Response · Authors · 2024-11-25
> > **Response to reviewer etKh**
> >
> > Thank you for your valuable feedback and for improving your score based on our results and responses. We understand your concerns regarding the reproducibility of our experiments and to address this, we have updated our anonymous source code repo (https://anonymous.4open.science/r/democratic_training-EB5A). We hope this additional resource will clarify any remaining questions and facilitate reproducibility.
> >
> > We appreciate your thoughtful review and the opportunity to strengthen the transparency and robustness of our work. If you encounter any issues with the code or require further clarification, we would be happy to assist.

---

### Official Review · Reviewer_hAa5 · 2024-10-27

**Soundness:** 3
**Presentation:** 3
**Contribution:** 2
**Rating:** 6
**Confidence:** 3

**Summary:**

To improve neural network robustness to targeted UAPs, this paper proposed an adversarial training-like method that fine-tunes a pretrained model to reduce middle-layer entropy.

**Strengths:**

1. The experiments are comprehensive.
2. The proposed defense is attack-agnostic which is more practical and efficient.
3. The proposed defense largely reduced the targeted attack success rate.

I tend to accept this pape. However, since I'm not familiar with UAP attack and defense baseline methods, I will listen to other reviewers and public comments and then decide.

**Weaknesses:**

1. UAP attacks evaluated in the paper were published in 2018,2019,2020 and seem out-of-date.
2. After democracy training, there is still a gap between ``AAcc.'' and clean accuracy. I wonder about the effectiveness of democracy training against non-targeted UAPs.
3. Average results in Table 4\&5 are ambiguous since there can be a large bias among different networks.

**Questions:**

1. For RQ3: did you adversarially train a model from scratch or just fine-tune a pretrained model with an adversarial training objective?
2. I'm not very convinced by the claim in Lines 239-243 that says middle-layer feature entropy suggests the model's classification confidence.

---

> ### Author Response · Authors · 2024-11-22
> **Response to Reviewer  hAa5 I**
>
> Thank you for your thorough review and valuable insights. We have provided detailed responses to your comments below.
> Response to weaknesses:
> 1. Thanks for the comments and we are glad to evaluate more recent UAP attacks. For the time being, we evaluated Democratic Training against SGA [4] published in 2023 and below shows the result:
> | | before || after ||
> | ------- | ----- | ----- | ----- | ----- |
> | model   | Aacc  | SR    | Aacc  | SR    |
> | NN1     | 0.133 | 0.722 | 0.592 | 0.005 |
> | NN2     | 0.067 | 0.806 | 0.415 | 0.096 |
> | NN3     | 0.147 | 0.641 | 0.510 | 0.011 |
> | NN4     | 0.034 | 0.999 | 0.904 | 0.009 |
> | NN5     | 0.107 | 0.798 | 0.743 | 0.020 |
> | NN6     | 0.227 | 0.776 | 0.812 | 0.034 |
> | average | 0.119 | 0.790 | 0.663 | 0.029 |
>
> The results show that Democratic Training is able to reduce the attack success rate from 81% to 2.9% and model accuracy is maintained high (>71%), which is consistent with the results reported in the draft. We will add these new results.

---

> ### Author Response · Authors · 2024-11-22
> **Response to Reviewer hAa5 II**
>
> 2. Thanks for the insightful comments. In this work, we focus on targeted UAP attacks which are both more relevant from an attacker point of view (i.e., so that the attacker can trigger specific target outcomes) and more challenging from a defender point of view. Our empirical study in section 3.2 shows the abnormal entropy spectrum caused by targeted UAPs and Democratic Training is designed accordingly. We have now managed to evaluate Democratic Training on non-targeted attacks as well and the results are summarized below:
> | | before ||| after |||
> | ----- | ----- | ----- | ----- | ----- | ----- | ----- |
> | model | cacc  | aacc  | SR    | cacc  | aacc  | SR    |
> | NN1   | 0.752 | 0.057 | 0.939 | 0.705 | 0.594 | 0.267 |
> | NN2   | 0.717 | 0.056 | 0.943 | 0.651 | 0.369 | 0.559 |
> | NN3   | 0.685 | 0.098 | 0.888 | 0.65  | 0.469 | 0.408 |
> | NN4   | 0.999 | 0.002 | 0.981 | 0.968 | 0.918 | 0.066 |
> | NN5   | 0.858 | 0.053 | 0.958 | 0.839 | 0.607 | 0.374 |
> | NN6   | 0.892 | 0.289 | 0.737 | 0.884 | 0.801 | 0.129 |
> | avg   | 0.817 | 0.093 | 0.908 | 0.783 | 0.626 | 0.301 |
>
> As we can see from the above table, although not designed for non-targeted UAPS, Democratic Training still reduces the attack SR from over 90% to 30% on average. This is indeed not as effective as targeted UAP defense performance and we believe this is due to the different entropy spectrum caused by the two types of UAPs. We also analyzed the entropy spectrum of clean and UAP perturbed samples and no clear separation of the two is observed (we will add the plot to the revised version).
> Hence, although non-targeted UAPs does not cause severe entropy change, enhancing a given model with low-entropy samples still improves the robustness against such perturbations to a certain level.
>
> 3. Thanks for the comments and the details of each network in Table 4 is shown below:
>
> | Setting      | Model | SR    | Aacc  | Delta CACC |
> | ------------ | ----- | ----- | ----- | ---------- |
> | Targeted     | NN1   | 0.239 | 0.441 | \-0.010    |
> |              | NN2   | 0.397 | 0.099 | \-0.010    |
> |              | NN3   | 0.069 | 0.480 | \-0.096    |
> |              | NN4   | 0.088 | 0.621 | \-0.067    |
> |              | NN5   | 0.018 | 0.622 | \-0.079    |
> |              | NN6   | 0.188 | 0.521 | \-0.359    |
> |              | avg   | 0.167 | 0.464 | \-0.104    |
> | Non-targeted | NN1   | 0.306 | 0.297 | \-0.063    |
> |              | NN2   | 0.655 | 0.271 | \-0.050    |
> |              | NN3   | 0.421 | 0.238 | \-0.160    |
> |              | NN4   | 0.499 | 0.305 | \-0.010    |
> |              | NN5   | 0.428 | 0.216 | \-0.289    |
> |              | NN6   | 0.422 | 0.445 | \-0.435    |
> |              | avg   | 0.455 | 0.295 | \-0.168    |
> | Known UAP    | NN1   | 0.000 | 0.554 | \-0.001    |
> |              | NN2   | 0.422 | 0.101 | 0.004      |
> |              | NN3   | 0.128 | 0.414 | \-0.014    |
> |              | NN4   | 0.163 | 0.649 | 0.004      |
> |              | NN5   | 0.005 | 0.750 | 0.004      |
> |              | NN6   | 0.619 | 0.385 | 0.000      |
> |              | avg   | 0.223 | 0.476 | 0.000      |
>
> For comparison with existing works. We compare the performance of CFN and FNS against Democratic Training on NN1, NN2 and NN3. Instead of testing different combinations of clr and dr setting, we adopt the recommended value in the original paper. The table below shows the details for each model. For SFR, as the method is not fully open-sourced, we can only evaluate the performance of GoogleNet trained on ImageNet dataset where a pretrained defense model is provided (Table 5 shows the result on GoogleNet).
>
> | | before || after (CFN) ||| after (FNS) |||
> | ----- | ------ | ------ | ----- | ----- | --------- | ----- | ----- | --------- |
> | model | Aacc   | SR     | Aacc  | SR    | Cacc drop | Aacc  | SR    | Cacc drop |
> | NN1   | 13.356 | 71.363 | 0.150 | 0.575 | 0.131     | 0.155 | 0.674 | 0.019     |
> | NN2   | 6.625  | 69.781 | 0.075 | 0.661 | 0.031     | 0.074 | 0.677 | 0.002     |
> | NN3   | 19.475 | 58.375 | 0.224 | 0.443 | 0.056     | 0.219 | 0.520 | 0.017     |

---

> > ### Comment · Reviewer_hAa5 · 2024-11-24
> > **Question about non-targeted attack**
> >
> > In the context of non-targeted attacks, why isn’t the adversarial accuracy complementary to the attack success rate, i.e., why doesn’t Aacc+SR=1?

---

> > > ### Author Response · Authors · 2024-11-25
> > > **Response to questions about non-targeted attack**
> > >
> > > Thanks for the comment and in this work we report fooling rate (FR) as the attack success rate (SR) following existing works [12, 13]. For non-target attack, fooling rate is used to measure the ratio of samples changing their prediction when UAP is added.  Given a test dataset $X$,a target model $f$ and a UAP $\delta$,
> > > For non-targeted attack:
> > >
> > > $SR = nFR = \sum_{x\in X}\frac{|f(x + \delta) \neq f(x)|}{|X|}$
> > >
> > > For targeted attack (let $t$ represent the target class, $X_t$ represent the samples with label $t$ in $X$), fooling ratio is used to measure the ratio of samples (not belonging to the target class) classified into the attack target class when UAP is added:
> > >
> > > $SR = tFR = \sum_{x\in (X - X_t)}\frac{|f(x + \delta) = t|}{|X| - |X_t|}$
> > >
> > > Hence, it is possible that $AAcc + SR \neq 1 $. Sorry about not defining the matrix clearly and we will add the definition in the revised paper accordingly.
> > >
> > > Reference:
> > >
> > > [12] Zhang, Chaoning, et al. "Understanding adversarial examples from the mutual influence of images and perturbations." Proceedings of the IEEE/CVF Conference on Computer Vision and Pattern Recognition. 2020
> > >
> > > [13]Weng, Juanjuan, et al. "Comparative evaluation of recent universal adversarial perturbations in image classification." Computers & Security 136 (2024): 103576.

---

> > > > ### Comment · Reviewer_hAa5 · 2024-11-25
> > > >
> > > > Thanks for your additional results. I will maintain my score and increase my confidence in this positive rating.

---

> > > > > ### Author Response · Authors · 2024-11-25
> > > > > **Response to reviewer hAa5**
> > > > >
> > > > > Thanks for your encouraging feedback and for increasing your confidence in the positive rating. We are pleased that the additional results have reinforced your evaluation, and we greatly appreciate your thoughtful review and support of our work.

---

> ### Author Response · Authors · 2024-11-22
> **Response to Reviewer hAa5 III**
>
> Response to questions:
> 1. In RQ3, we aim to compare the efficiency of adversarial examples and our low-entropy examples in mitigating the effect of UAPs when finetuning a given model. Moreover, we conduct an additional experiment to evaluate  adversarial training from scratch over UAP attacks. We compare the UAP defense performance of TRADES[1] (which is widely recognized adversarial training method) and ours on a wideresent model trained on cifar-10 dataset. The result is summarized below:
>
> | | before || after (ours) || after (TRADES) ||
> | ------ | ----- | ----- | ----- | ----- | ----- | ----- |
> | target | Aacc. | SR    | Aacc. | SR    | Aacc. | SR    |
> | 0      | 0.155 | 0.922 | 0.871 | 0.009 | 0.819 | 0.018 |
> | 1      | 0.144 | 0.927 | 0.864 | 0.016 | 0.816 | 0.004 |
> | 2      | 0.143 | 0.954 | 0.857 | 0.050 | 0.816 | 0.028 |
> | 3      | 0.116 | 0.972 | 0.875 | 0.046 | 0.819 | 0.042 |
> | 4      | 0.148 | 0.938 | 0.874 | 0.027 | 0.818 | 0.015 |
> | 5      | 0.226 | 0.861 | 0.861 | 0.036 | 0.818 | 0.042 |
> | 6      | 0.139 | 0.963 | 0.860 | 0.048 | 0.811 | 0.036 |
> | 7      | 0.143 | 0.942 | 0.849 | 0.045 | 0.812 | 0.010 |
> | 8      | 0.171 | 0.928 | 0.852 | 0.047 | 0.815 | 0.015 |
> | 9      | 0.134 | 0.954 | 0.847 | 0.014 | 0.817 | 0.010 |
> | avg    | 0.152 | 0.936 | 0.861 | 0.034 | 0.816 | 0.022 |
> | | Clean Acc: 0.931 || Clean Acc: 0.901 || Clean Acc: 0.827 ||
>
> Based on the above result, both TRADES and Democratic Training are effective in mitigating the effect of UAPs. However, TRADES sacrifices model accuracy for over 10% while Democratic Training suffers from much less reduction (3%). Furthermore, Democratic Training repairs the model within 20 min while TRADES takes over 15 hrs to train a robust model on the same machine. Hence, our method is effective in UAP defense and is much more time efficient.
>
> 2. Thanks for the comments. In this work, we would like to use entropy to characterise how uncertain the model is on the classification of the intermediate features. Higher entropy suggests the features are ambiguous while lower entropy indicates the model is more certain on classifying the features. Based on our empirical study, UAPs will cause the layer-wise entropy to drop, and such lower entropy indicates the model is more certain on its classification at the same layer. Existing work [12] shows that UAPs contain dominant features over original image and we argue that such dominant features cause the layer-wise entropy to drop which dominates the model prediction. We will improve our presentation accordingly in the revised version.
>
> References
>
> [1] Costa, J. C., Roxo, T., Proença, H., & Inácio, P. R. (2024). How deep learning sees the world: A survey on adversarial attacks & defenses. IEEE Access.
>
> [12] Zhang, Chaoning, et al. "Understanding adversarial examples from the mutual influence of images and perturbations." Proceedings of the IEEE/CVF Conference on Computer Vision and Pattern Recognition. 2020

---

### Official Review · Reviewer_1Bzp · 2024-11-01

**Soundness:** 3
**Presentation:** 3
**Contribution:** 3
**Rating:** 6
**Confidence:** 4

**Summary:**

The paper presents a novel defense method called Democratic Training to mitigate the impact of Universal Adversarial Perturbations (UAPs) on deep neural networks.  The authors observed that UAPs lead to an abnormal entropy spectrum in hidden layers, which shows the model's prediction is dominated by a small subset of features. Democratic Training mitigates this issue by increasing entropy to ensure model predictions rely on a wider range of features. The approach was evaluated on multiple models and datasets and was found to be effective in reducing attack success rates while preserving accuracy on clean data.

**Strengths:**

1. The use of entropy to reveal the dominance of UAPs and the concept of Democratic Training as a defense mechanism is innovative.
2. The method was evaluated across various neural network architectures and benchmark datasets, which strengthens the claim of its general applicability.
3. Unlike other defense methods, Democratic Training does not require architectural modifications, which makes it easy to integrate into existing systems

**Weaknesses:**

1. The evaluation focused primarily on benchmark datasets and common UAP generation methods. It would be beneficial to see how this approach performs on more sophisticated and adaptive attacks, such as adversarial examples generated in dynamic environments.
2. The proposed method mainly works well on CNN. Authors should validate it in more types of networks, such as transformers.
3.  The method requires access to a small set of clean data for entropy measurement and training, which might not always be practical

**Questions:**

1. How about performances of the method on ViT?
2. What's the time cost of the method?

---

> ### Author Response · Authors · 2024-11-22
> **Response to Reviewer 1Bzp**
>
> We greatly appreciate your comprehensive review and thoughtful suggestions. Please find our detailed responses to your comments below.
> Response to weekness:
> 1. Thanks for the comments and we have evaluated Democratic Training on adaptative attacks. The results are summarized below and are also available in Appendix 7.4 ADAPTIVE ATTACKS . In summary, although secondary UAP attacks on Democratic Training repaired models can still generate UAPs that successfully fool the models, our defense keeps the secondary attack success rate to a very low level while keeping the adversarial accuracy high.
>
> | Model | AAcc  | SR    |
> | ----- | ----- | ----- |
> | NN1   | 0.418 | 0.210 |
> | NN2   | 0.303 | 0.249 |
> | NN3   | 0.437 | 0.107 |
> | NN4   | 0.881 | 0.037 |
> | NN5   | 0.445 | 0.307 |
> | NN6   | 0.545 | 0.405 |
>
> Furthermore, we consider more advanced adversaries who may tailor the adversarial examples trying to bypass democratic training. To explore Democratic Training’s resilience to such attackers, we conduct experiments such that when generating UAP,  the attacker further control the change in layer-wise entropy. Based on DF-UAP, the optimisation loss function is modified as $L(i) = (1 - weight) * L_{cce}(i, target) - weight *  H_l(i)$, where $i$ represents a clean training inputs, target represents the attack target class and $H_l(i)$ represents the layer-wise entropy loss. We use $H_l(i)$ to control the entropy change caused by the UAP and parameter $weight$ is used to control the importance of $H_l(i)$ over attack success rate. We conduct such advanced attack on all models with $weight$ set to 0.1~0.9 and and all models how similar results. For illustration purpose, results on $NN_1$ are summarised below:
>
> |       | before || after ||
> | ----- | ------ | ----- |----- |----- |
> | alpha | Aacc   | SR    | Aacc | SR |
> | 0.0   | 0.118  | 0.764 | 0.619 | 0.001 |
> | 0.1   | 0.121  | 0.775 | 0.619 | 0.001 |
> | 0.2   | 0.118  | 0.759 | 0.619 | 0.001 |
> | 0.3   | 0.128  | 0.764 | 0.613 | 0.001 |
> | 0.4   | 0.127  | 0.761 | 0.612 | 0.001 |
> | 0.5   | 0.125  | 0.759 | 0.608 | 0.000 |
> | 0.6   | 0.141  | 0.745 | 0.618 | 0.000 |
> | 0.7   | 0.161  | 0.693 | 0.599 | 0.001 |
> | 0.8   | 0.185  | 0.657 | 0.609 | 0.001 |
> | 0.9   | 0.207  | 0.568 | 0.628 | 0.000 |
>
> Based on above result, increasing en_weight will cause the attack performance to drop, i.e., the attack success rate starts to drop when $weight$ > 0.5 and the attack SR is below 60% when en_weight is set to 0.9. Our defense stays effective across different en_weight settings where the attack SR is recued to <1% for all scenarios. We observe similar results on other models as well. Hence, knowing how Democratic Training enhance the model and control the change in layer-wise entropy during attack process, the adversary is still not able to bypass our defense effectively.
>
> 2. Thanks for the recommendation and we are glad to extend our work to more types of networks in our future work. Sorry to say that due to the time constraint, we haven’t been able to finish experiments on transformers yet.
>
> 3. Thanks for the comments and we agree that it might now always be the case that a clean dataset is available. In our problem definition, our aim is to protect a model trained by a third party. In this scenario, a small set of clean data is usually available for testing and validation purposes. We believe such assumption is reasonable in practice and is usually made in existing works on defense against neural network attacks [5,6,7,8,9,10,11]. Having said that, we are glad to explore data-free defense methods against UAPs in our future works (for instance, using a synthetic clean dataset).

---

> ### Author Response · Authors · 2024-11-22
> **Response to Reviewer 1Bzp II**
>
> Response to questions:
> 1. We are glad to extend our method on ViT and other neural network architectures in our future works. Sorry to say that due to the time constraint, we haven’t been able to finish experiments on transformers yet.
>
> 2. Sorry for omitting the information. The time cost of enhancing all models are listed below (which we will add in the draft).
>
> | model | time  |
> | ----- | ----- |
> | NN1   | 8min  |
> | NN2   | 36min |
> | NN3   | 7min  |
> | NN4   | 18min |
> | NN5   | 30min |
> | NN6   | 16min |
>
> References
>
> [5] Shen, Guangyu, et al. "Backdoor scanning for deep neural networks through k-arm optimization." International Conference on Machine Learning. PMLR, 2021.
>
> [6] Tang, Di, et al. "Demon in the variant: Statistical analysis of {DNNs} for robust backdoor contamination detection." 30th USENIX Security Symposium (USENIX Security 21). 2021.
>
> [7] Liu, Yingqi, et al. "Complex backdoor detection by symmetric feature differencing." Proceedings of the IEEE/CVF Conference on Computer Vision and Pattern Recognition. 2022.
>
> [8] Wu, Dongxian, and Yisen Wang. "Adversarial neuron pruning purifies backdoored deep models." Advances in Neural Information Processing Systems 34 (2021): 16913-16925.
>
> [9] Ho, Chih-Hui, and Nuno Vasconcelos. "DISCO: Adversarial defense with local implicit functions." Advances in Neural Information Processing Systems 35 (2022): 23818-23837.
>
> [10] Akhtar, Naveed, Jian Liu, and Ajmal Mian. "Defense against universal adversarial perturbations." Proceedings of the IEEE conference on computer vision and pattern recognition. 2018.
>
> [11] Borkar, Tejas, Felix Heide, and Lina Karam. "Defending against universal attacks through selective feature regeneration." Proceedings of the IEEE/CVF conference on computer vision and pattern recognition. 2020.

---

> > ### Comment · Reviewer_xSaF · 2024-11-23
> >
> > I would not mind the time spent for the defense too much, as most adversarial training methods use massive amounts of data. However, it is important to report your device details if your want to report this point.

---

> > > ### Author Response · Authors · 2024-11-23
> > >
> > > Dear Reviewer xSaF,
> > >
> > > Thanks for the recommendation and the machine we used to run all reported experiments is with 96-Core 1.4GHz CPU and 60GB system memory with an NVIDIA 24GB RTX 4090 GPU. We mentioned this in the original paper Section 4 but understand that it will be much clearer if we have provided this information in above comments.

---

> ### Comment · Reviewer_1Bzp · 2024-11-26
> **Response after rebuttal**
>
> Thanks for the response. It solves my concerns. I'd like to update the score.

---

> > ### Author Response · Authors · 2024-11-27
> >
> > Dear Reviewer 1Bzp,
> >
> > Thanks for your kind feedback and for taking the time to review our responses. We truly appreciate your thoughtful engagement and are glad that our clarifications have addressed your concerns.

---

### Official Review · Reviewer_xSaF · 2024-11-02

**Soundness:** 3
**Presentation:** 3
**Contribution:** 3
**Rating:** 8
**Confidence:** 5

**Summary:**

The paper presents an investigation into the defense against universal adversarial perturbations (UAPs), with a particular focus on targeted UAPs. The authors have made a notable observation regarding the entropy spectrum in hidden layers when UAPs are introduced, which serves as the cornerstone for their proposed 'democratic training' approach. This method aims to enhance the adversarial robustness of neural network models. The empirical results provided in the paper demonstrate the efficacy of the approach, as well as its ability to maintain the clean accuracy of the model.

In general, this paper is well-structured and presents a novel approach, which meets the basic criteria of ICLR conference. However, there are some aspects that could be improved or expanded upon to enhance the overall quality and impact of the paper.

**Strengths:**

1. This paper is well-written and easy-to-follow.
2. The paper makes a commendable observation concerning the entropy spectrum in deep neural network layers, which is a significant contribution to the field and forms the basis for the proposed defense mechanism.
3. The efficiency of the proposed democratic training method is noteworthy. It circumvents the need to generate UAPs during training, instead utilizing a limited number of epochs to identify low-entropy examples, which is a resourceful approach.

**Weaknesses:**

1. The threat model employed in the experiments primarily utilizes gradient-based attack methods. These methods presuppose access to the model's parameters, aligning with white-box attack scenarios. This appears to be at odds with the assertion in Section 2.3 that adversarial knowledge does not extend to the internal parameters of the model. Clarification on this point would be beneficial.

2. The comparison with adversarial training methods may require further refinement. Adversarial training aims to bolster adversarial accuracy by integrating adversarial examples with clean examples during training. Constraining the number of training epochs could result in an underfit model, which may not provide a fair benchmark. Additionally, it would be advantageous to include a comparison with the widely recognized TRADES method[1], which is absent from the current manuscript.

3. The potential for adaptive attacks warrants consideration. If adversaries are aware of the defense strategy, they could tailor adversarial examples to bypass the defense. I know that in the threat model, no adaptive attacks are considered since the attackers do not know the internal details of the models. However, the chosen attack methods in the experiments inherently rely on gradient information. So I would suggest that the authors should consider the potential for adaptive attacks.

4. The scope of the experiments is largely limited to datasets comprising natural images. It would be beneficial to extend the evaluation to smaller-scale datasets, such as CIFAR-10, to complement the findings and potentially leverage open-source robust models for further exploration of the neuron entropy spectrum concept.

5. While the paper discusses various existing defensive methods against UAPs and includes experimental comparisons, a direct comparison with state-of-the-art methods is missing. It is recommended to condense the background section and incorporate a more thorough comparison with leading-edge techniques.

6. Minor Issues
  (1) Please consider to reduce the margin between Figure 1 and the text.
  (2) Suggesting to add necessary notations (SR) from the main test to the Table 2 for better understanding.


[1] Zhang, Hongyang, et al. "Theoretically principled trade-off between robustness and accuracy." International conference on machine learning. PMLR, 2019.

**Questions:**

See weakness.

---

> ### Author Response · Authors · 2024-11-22
> **Response to Reviewer xSaF**
>
> We deeply appreciate your thorough evaluation and insightful feedback. Below, we provide our detailed responses to the points raised.
> Response to weaknesses:
> 1. Thanks for the comments and sorry for the mistake. Indeed, our approach is designed to defend against a strong adversary who can conduct UAP attacks with white-box access to the model. We have modified our threat model in the revised paper accordingly.
> 2. Thanks for the insightful comment. In RQ3, we aim to compare the efficiency of adversarial examples and our low-entropy examples in mitigating the effect of UAPs when finetuning a given model. We are happy to evaluate ‘full’ adversarial training on UAPs and compare the result with ours. Given the time constraints, we have now evaluated TRADES [2] on a wideresent model trained on cifar-10 dataset and compared the result with ours. The results are summarised below.
>
> |        | before            | after (ours)      | after (TRADES)   |
> | ------ | ----------------- | ----------------- | ---------------- |
> | target | Aacc.             | SR                | Aacc.            | SR | Aacc. | SR |
> | 0      | 0.155             | 0.922             | 0.871            | 0.009 | 0.819 | 0.018 |
> | 1      | 0.144             | 0.927             | 0.864            | 0.016 | 0.816 | 0.004 |
> | 2      | 0.143             | 0.954             | 0.857            | 0.050 | 0.816 | 0.028 |
> | 3      | 0.116             | 0.972             | 0.875            | 0.046 | 0.819 | 0.042 |
> | 4      | 0.148             | 0.938             | 0.874            | 0.027 | 0.818 | 0.015 |
> | 5      | 0.226             | 0.861             | 0.861            | 0.036 | 0.818 | 0.042 |
> | 6      | 0.139             | 0.963             | 0.860            | 0.048 | 0.811 | 0.036 |
> | 7      | 0.143             | 0.942             | 0.849            | 0.045 | 0.812 | 0.010 |
> | 8      | 0.171             | 0.928             | 0.852            | 0.047 | 0.815 | 0.015 |
> | 9      | 0.134             | 0.954             | 0.847            | 0.014 | 0.817 | 0.010 |
> | avg    | 0.152             | 0.936             | 0.861            | 0.034 | 0.816 | 0.022 |
> |        | Clean Acc.: 0.931 | Clean Acc.: 0.901 | Clean Acc: 0.827 |
>
> Based on the above result, both TRADES and Democratic Training are effective in mitigating the effect of UAPs. However, TRADES sacrifices model accuracy for over 10% while Democratic Training suffers from much less reduction (3%). Furthermore, Democratic Training repairs the model within 20 min while TRADES takes over 15 hrs to train a robust model on the same machine. Hence, our method is effective in UAP defense and is much more time efficient.
>
> 3. Thanks for the recommendation and we agree that advanced adversaries may tailor the adversarial examples trying to bypass democratic training. To explore Democratic Training’s resilience to such attackers, we conduct experiments such that when generating UAP,  the attacker further control the change in layer-wise entropy. Based on DF-UAP, the optimisation loss function is modified as $L(i) = (1 - weight) * L_{cce}(i, target) - weight *  H_l(i)$, where $i$ represents a clean training inputs, target represents the attack target class and $H_l(i)$ represents the layer-wise entropy loss. We use $H_l(i)$ to control the entropy change caused by the UAP and parameter $weight$ is used to control the importance of $H_l(i)$ over attack success rate. We conduct such advanced attack on all models with $weight$ set to 0.1~0.9 and and all models how similar results. For illustration purpose, results on $NN_1$ are summarised below:
>
> |       | before || after ||
> | ----- | ------ | ----- |----- |----- |
> | alpha | Aacc   | SR    | Aacc | SR |
> | 0.0   | 0.118  | 0.764 | 0.619 | 0.001 |
> | 0.1   | 0.121  | 0.775 | 0.619 | 0.001 |
> | 0.2   | 0.118  | 0.759 | 0.619 | 0.001 |
> | 0.3   | 0.128  | 0.764 | 0.613 | 0.001 |
> | 0.4   | 0.127  | 0.761 | 0.612 | 0.001 |
> | 0.5   | 0.125  | 0.759 | 0.608 | 0.000 |
> | 0.6   | 0.141  | 0.745 | 0.618 | 0.000 |
> | 0.7   | 0.161  | 0.693 | 0.599 | 0.001 |
> | 0.8   | 0.185  | 0.657 | 0.609 | 0.001 |
> | 0.9   | 0.207  | 0.568 | 0.628 | 0.000 |
>
> Based on above result, increasing en_weight will cause the attack performance to drop, i.e., the attack success rate starts to drop when $weight$ > 0.5 and the attack SR is below 60% when en_weight is set to 0.9. Our defense stays effective across different en_weight settings where the attack SR is recued to <1% for all scenarios. We observe similar results on other models as well. Hence, knowing how Democratic Training enhance the model and control the change in layer-wise entropy during attack process, the adversary is still not able to bypass our defense effectively.

---

> ### Author Response · Authors · 2024-11-22
> **Response to Reviewer xSaF II**
>
> 4. Thanks for the comment. We are glad to evaluate Democratic Training on smaller-scale datasets. Below is the performance on a wideresenet model rained on cifar-10. It can be observed that the results are consistent with what we reported. We will add the details in the draft.
>
> |        | before || after ||
> | ---- | ---- | ---- | ---- | ---- |
> | target | Aacc. | SR | Aacc. | SR |
> | 0      | 0.155 | 0.922 | 0.871 | 0.009 |
> | 1      | 0.144 | 0.927 | 0.864 | 0.016 |
> | 2      | 0.143 | 0.954 | 0.857 | 0.050 |
> | 3      | 0.116 | 0.972 | 0.875 | 0.046 |
> | 4      | 0.148 | 0.938 | 0.874 | 0.027 |
> | 5      | 0.226 | 0.861 | 0.861 | 0.036 |
> | 6      | 0.139 | 0.963 | 0.860 | 0.048 |
> | 7      | 0.143 | 0.942 | 0.849 | 0.045 |
> | 8      | 0.171 | 0.928 | 0.852 | 0.047 |
> | 9      | 0.134 | 0.954 | 0.847 | 0.014 |
> | avg    | 0.152 | 0.936 | 0.861 | 0.034 |
> |        | Clean Acc.: 0.931 | Clean Acc.: 0.901 |
>
> 5. Thanks for the comments, we will improve our background section in the revised version. The three existing methods we are comparing with are designed for UAP defenses. We are glad to extend the comparison against more general adversarial defenses. Given the time constraint, we have now managed to compare with two additional two defense methods. That is, we evaluated TRADES against UAPs and compared the result with Democratic Training. Furthermore, we evaluated the performance of DensePure [3] on the wideresenet trained on cifar-10 the the performance comparison is shown below:
>
> |        | Before || Ours || DensePure||
> | ------ | ----- | ----- | ----- | ----- | ----- | ----- |
> | target | Aacc. | SR    | Aacc. | SR    | Aacc. | SR    |
> | 0      | 0.155 | 0.922 | 0.871 | 0.009 | 0.820 | 0.010 |
> | 1      | 0.144 | 0.927 | 0.864 | 0.016 | 0.790 | 0.000 |
> | 2      | 0.143 | 0.954 | 0.857 | 0.050 | 0.790 | 0.010 |
> | 3      | 0.116 | 0.972 | 0.875 | 0.046 | 0.810 | 0.040 |
> | 4      | 0.148 | 0.938 | 0.874 | 0.027 | 0.790 | 0.010 |
> | 5      | 0.226 | 0.861 | 0.861 | 0.036 | 0.810 | 0.010 |
> | 6      | 0.139 | 0.963 | 0.860 | 0.048 | 0.800 | 0.010 |
> | 7      | 0.143 | 0.942 | 0.849 | 0.045 | 0.790 | 0.000 |
> | 8      | 0.171 | 0.928 | 0.852 | 0.047 | 0.820 | 0.010 |
> | 9      | 0.134 | 0.954 | 0.847 | 0.014 | 0.800 | 0.000 |
> | avg    | 0.152 | 0.936 | 0.861 | 0.034 | 0.802 | 0.010 |
> |        | Clean Acc.: 0.931 || Clean Acc.: 0.901 || Clean Acc: 0.810 ||
>
> DensePure is effective in reducing the UAP attack success rate, but model accuracy is reduced by 12%. Democratic Training reduces the attack success rate to 3.4% and model accuracy is maintained high (3% reduction). Furthermore, as DensePure is a technique based on input sample purification, a large overhead is incurred for each inference. Based on our experiment, DensePure introduces about 560s overhead per inference (with the recommended setting).
>
> 6. We will fix these issues in the revised version accordingly.
>
>
> References
>
> [1] Costa, J. C., Roxo, T., Proença, H., & Inácio, P. R. (2024). How deep learning sees the world: A survey on adversarial attacks & defenses. IEEE Access.
>
> [2] Zhang, Hongyang, et al. "Theoretically principled trade-off between robustness and accuracy." International conference on machine learning. PMLR, 2019.
>
> [3] Chen, Zhongzhu, et al. "DensePure: Understanding Diffusion Models towards Adversarial Robustness." Workshop on Trustworthy and Socially Responsible Machine Learning, NeurIPS 2022. 2022.

---

> > ### Comment · Reviewer_xSaF · 2024-11-23
> > **Concerns Addressed**
> >
> > Dear Authors,
> >
> > Thank you for addressing the issues I mentioned earlier. It looks like you've covered most of them, which is great. If you add these changes to your final paper, I'll be happy to improve my rating. But I still have a few points to bring up:
> >
> > 1. **Comparison with TRADES:**
> >     1) Overall, the results look good.
> >     2) I noticed a small thing – did you mean to write "advacc" as "SR"? Could you please check this?
> >     3) The time it takes for TRADES and your method to work isn't a big deal since speed isn't usually a concern with white-box defense. Still, it's nice that you mentioned it. If you decide to include this in your paper, it would be helpful to tell us about the computer or device you used.
> >     4) Have you thought about combining your method with adversarial training like TRADES?
> >
> >
> > 2. **Adaptive Attack:** Your current test for the adaptive attack doesn't show the entropy values of your examples, which is important to know if the attack is truly adaptive. The results you've given does give us some idea of how well the attack works, though.

---

> ### Author Response · Authors · 2024-11-23
> **Response to additional comments of reviwer xSaF**
>
> Dear reviewer xSaF:
>
> Thanks for your comments and we will add these changes in our revised paper. If there are any specific areas where additional detail or explanation could further enhance the presentation, we would be happy to address them.
>
> Response to additional points:
>
> **Comparison with TRADES:**
> 1. Thanks for the positive assessment of our results and we appreciate your recognition of its overall quality.
> 2. Sorry about the mistake, and yes previously we report the advacc after defense for our method and TRADES. Below is the result on both Advacc and SR.
>
> | | before || after (ours) || after (TRADES) ||
> | ------ | ----- | ----- | ----- | ----- | ----- | ----- |
>  | target | Aacc. | SR    | Aacc. | SR    | Aacc. | SR    |
> | 0      | 0.155 | 0.922 | 0.871 | 0.009 | 0.819 | 0.018 |
> | 1      | 0.144 | 0.927 | 0.864 | 0.016 | 0.816 | 0.004 |
> | 2      | 0.143 | 0.954 | 0.857 | 0.050 | 0.816 | 0.028 |
> | 3      | 0.116 | 0.972 | 0.875 | 0.046 | 0.819 | 0.042 |
> | 4      | 0.148 | 0.938 | 0.874 | 0.027 | 0.818 | 0.015 |
> | 5      | 0.226 | 0.861 | 0.861 | 0.036 | 0.818 | 0.042 |
> | 6      | 0.139 | 0.963 | 0.860 | 0.048 | 0.811 | 0.036 |
> | 7      | 0.143 | 0.942 | 0.849 | 0.045 | 0.812 | 0.010 |
> | 8      | 0.171 | 0.928 | 0.852 | 0.047 | 0.815 | 0.015 |
> | 9      | 0.134 | 0.954 | 0.847 | 0.014 | 0.817 | 0.010 |
> | avg    | 0.152 | 0.936 | 0.861 | 0.034 | 0.816 | 0.022 |
> | | Clean Acc: 0.931 || Clean Acc: 0.901 || Clean Acc: 0.827 ||
>
> 3. Thanks for the recommendation and the machine we used to run all reported experiments is with 96-Core 1.4GHz CPU and 60GB system memory with an NVIDIA 24GB RTX 4090 GPU.
> 4. Thanks for the recommendation and we are glad to combine Democratic Training with adversarial training like TRADES. This may further improve the performance of Democratic Training and would likely to help extending Democratic Training to other types of adversarial attacks besides UAPs.
>
> **Adaptive Attack**
>
> Thanks for the comments and below table summarize the mean entropy of samples with and without UAPs generated with different $weight$ settings:
> | | before ||| after |||
> | ------ | ----- | ----- | ------- | ----- | ----- | ------- |
> | weight | Aacc  | SR    | entropy | Aacc  | SR    | entropy |
> | 0.0    | 0.118 | 0.764 | 5.62    | 0.619 | 0.001 | 7.39    |
> | 0.1    | 0.121 | 0.775 | 6.07    | 0.619 | 0.001 | 7.39    |
> | 0.2    | 0.118 | 0.759 | 6.29    | 0.619 | 0.001 | 7.40    |
> | 0.3    | 0.128 | 0.764 | 6.35    | 0.613 | 0.001 | 7.39    |
> | 0.4    | 0.127 | 0.761 | 6.89    | 0.612 | 0.001 | 7.40    |
> | 0.5    | 0.125 | 0.759 | 7.07    | 0.608 | 0.000 | 7.39    |
> | 0.6    | 0.141 | 0.745 | 7.13    | 0.618 | 0.000 | 7.39    |
> | 0.7    | 0.161 | 0.693 | 7.16    | 0.599 | 0.001 | 7.39    |
> | 0.8    | 0.185 | 0.657 | 7.33    | 0.609 | 0.001 | 7.40    |
> | 0.9    | 0.207 | 0.568 | 7.43    | 0.628 | 0.000 | 7.39    |
>
> *clean sample entropy is 7.1
>
> As we can see from above result, as $weight$ increases, the entropy value of samples with UAP also increases and the value exceeds that of clean sample when $weight >= 0.6$. It is also observed that the attack success rate starts to drop when $weight$ is greater than 0.6. With our enhanced model, the entropy value of samples with UAP stays high for different $weight$ settings and the attack success rate is kept below 1%. These results show that even when the advanced attacker tries to control the entropy drop when training an UAP, Democratic Training is still able to protect a given model effectively.

---

### Author Response · Authors · 2024-11-27
**Change Summary on Revised Paper**

Dear All,

We deeply appreciate your thorough evaluation and insightful feedback. We have updated our paper accordingly. Below is a summary of the changes:

1. Update Abstract: Democratic training is evaluated on one more model, one more benchmark dataset and one more UAP attack method
2. Add Section 2.2 Evaluation Metrics: add definitions of SR and AAcc.
3. Update Section 2.4 Threat Model: correct descriptions on adversarial capabilities and knowledge
4. Update Section 3.2 Entropy Analysis: improve descriptions on empirical study
5. Update Section 3.3 Entropy-based Repair: improve descriptions with symbols used in Algorithm 1, Algorithm 2 and Equation 6 properly introduced
6. Update Section 4.1 Experimental Setup: new dataset evaluatedL CIFAR-10; update Table 1
7. Update RQ1: add experimental results for $NN_7$ trained on CIFAR-10 with widsresnet architecture; update Table 2 and Figure 3
8. Update RQ2: add experimental results for UAP attack method SGA; update Table 3
9. Update RQ3: add experimental results for TRADES; update Table 4
10. Update RQ4: add experimental results for DensePure; update Table 5
11. Update Section 5 Related Works: improve background on defense against adversarial attacks
12. Add Future Works section in Appendix 7.1
13. Update Datasets Used in Our Experiments in Appendix 7.3: add descriptions on CIFAR-10
14. Update Adaptive Attacks in Appendix 7.5: add experimental results for advanced adversaries; add Table 8
15: Add Entropy Analysis on Other UAPs in Appendix 7.6: add Figure 4
16. Add Non-targeted UAP Attacks in Appendix 7.7: add results for non-targeted UAP defense performance; add Table 9; add Figure 5
17. Address comments on formatting



Thanks!

BR.
Authors

---

### Meta-Review · Area_Chair_67M6 · 2024-12-18

**Metareview:**

The paper presents a novel defense method called Democratic Training to mitigate the impact of Universal Adversarial Perturbations (UAPs) on deep neural networks. The authors observed that UAPs lead to an abnormal entropy spectrum in hidden layers, which shows the model's prediction is dominated by a small subset of features. Democratic Training mitigates this issue by increasing entropy to ensure model predictions rely on a wider range of features. The approach was evaluated on multiple models and datasets and was found to be effective in reducing attack success rates while preserving accuracy on clean data. There are some merits in this paper. For example, the use of entropy to reveal the dominance of UAPs and the concept of Democratic Training as a defense mechanism is innovative. The method was evaluated across various neural network architectures and benchmark datasets, which strengthens the claim of its general applicability. Unlike other defense methods, Democratic Training does not require architectural modifications, which makes it easy to integrate into existing systems. Moreover, this paper is well-written and easy-to-follow. The paper makes a commendable observation concerning the entropy spectrum in deep neural network layers, which is a significant contribution to the field and forms the basis for the proposed defense mechanism. The efficiency of the proposed democratic training method is noteworthy. It circumvents the need to generate UAPs during training, instead utilizing a limited number of epochs to identify low-entropy examples, which is a resourceful approach. While the reviewers had some concerns about reproducibility, the authors did a particularly good job in their rebuttal. Therefore, all of us have agreed to accept this paper for publication! Please include the additional discussion in the next version.

**Additional Comments On Reviewer Discussion:**

Some reviewers raise the score after the rebuttal.

---

### Decision · Program_Chairs · 2025-01-22

Accept (Poster)